# LEARNING TO MULTI-TASK BY ACTIVE SAMPLING

**Sahil Sharma***
Department of Computer Science and Engineering
Indian Institute of Technology, Madras

**Ashutosh Kumar Jha***
Department of Mechanical Engineering
Indian Institute of Technology, Madras

**Parikshit S Hegde**
Department of Electrical Engineering
Indian Institute of Technology, Madras

**Balaraman Ravindran**
Department of Computer Science and Engineering
and Robert Bosch Centre for Data Science
and AI (RBC-DSAI)
Indian Institute of Technology, Madras

## ABSTRACT

One of the long-standing challenges in Artificial Intelligence for learning goal-directed behavior is to build a *single agent* which can solve multiple tasks. Recent progress in multi-task learning for goal-directed sequential problems has been in the form of *distillation based learning* wherein a student network learns from multiple task-specific expert networks by mimicking the task-specific policies of the expert networks. While such approaches offer a promising solution to the multi-task learning problem, they require supervision from large expert networks which require extensive data and computation time for training. In this work, we propose an efficient multi-task learning framework which solves multiple goal-directed tasks in an *on-line* setup without the need for expert supervision. Our work uses active learning principles to achieve multi-task learning by sampling the *harder* tasks more than the easier ones. We propose three distinct models under our active sampling framework. An adaptive method with extremely competitive multi-tasking performance. A UCB-based meta-learner which casts the problem of picking the next task to train on as a multi-armed bandit problem. A meta-learning method that casts the next-task picking problem as a full Reinforcement Learning problem and uses actor critic methods for optimizing the multi-tasking performance directly. We demonstrate results in the Atari 2600 domain on seven multi-tasking instances: three 6-task instances, one 8-task instance, two 12-task instances and one 21-task instance.

## 1 INTRODUCTION

Deep Reinforcement Learning (DRL) arises from the combination of the representation power of Deep learning (DL) (LeCun et al., 2015; Bengio et al., 2009) with the use of Reinforcement Learning (RL) (Sutton & Barto, 1998) objective functions. DRL agents can solve complex visual control tasks directly from raw pixels (Guo et al., 2014; Mnih et al., 2015; Schulman et al., 2015; Lillicrap et al., 2015; Schaul et al., 2015; Mnih et al., 2016a; Van Hasselt et al., 2016; Vezhnevets et al., 2016; Bacon et al., 2017; Sharma et al., 2017; Jaderberg et al., 2017).
However, models trained using such algorithms tend to be task-specific because they train a different network for different tasks, however similar the tasks are. This inability of the AI agents to generalize across tasks motivates the field of multi-task learning which seeks to find a *single* agent (in the case of DRL algorithms, a *single* deep neural network) which can perform well on *all* the tasks. Training a neural network with a multi-task learning (MTL) algorithm on any fixed set of tasks (which we call a *multi tasking instance* (MTI)) leads to an instantiation of a multi-tasking agent (MTA) (we use the terms Multi-Tasking Network (MTN) and MTA interchangeably). Such an MTA would possess the ability to learn task-agnostic representations and thus generalize learning across different tasks.
Successful DRL approaches to the goal-directed MTL problem fall into two categories. First, there are approaches that seek to extract the prowess of multiple task-specific expert networks into a

---

* - The authors had equal contribution

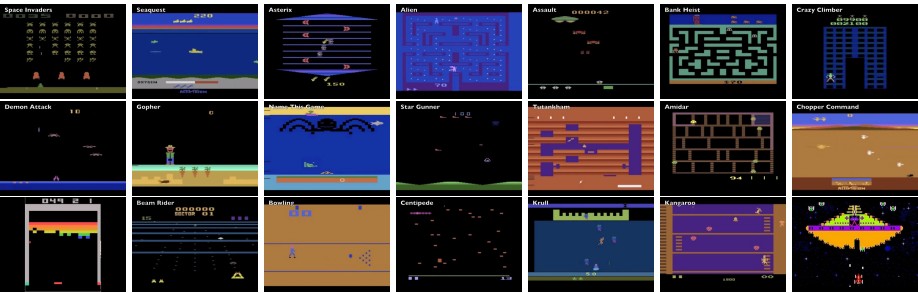

Figure 1: Multi Tasking Instance MT7 (21 tasks). A playlist of game-play for MT7 is at: `https://goo.gl/GBXfWD`. It verifies our claim; tasks are visually different & difficult.

single student network. The Policy Distillation framework (Rusu et al., 2016a) and Actor-Mimic Networks (Parisotto et al., 2016) fall into this category. These works train $k$ task-specific expert networks (DQNs (Mnih et al., 2015)) and then *distill* the individual task-specific policies learned by the expert networks into a single student network which is trained using supervised learning. While these approaches eventually produce a single a network that solves multiple tasks, individual expert networks must first be trained, and this training tends to be extremely computation and data intensive.

The second set of DRL approaches to multi-tasking are related to the field of transfer learning. Many recent DRL works (Parisotto et al., 2016; Rusu et al., 2016b; Rajendran et al., 2017; Fernando et al., 2017) attempt to solve the transfer learning problem. Progressive networks (Rusu et al., 2016b) is one such framework which can be adapted to the MTL problem. Progressive networks iteratively learn to solve each successive task that is presented. Thus, they are not a truly on-line learning algorithm. Progressive Networks instantiate a task-specific *column network* for each new task. This implies that the number of parameters they require grows as a large linear factor with each new task. This limits the scalability of the approach with the results presented in the work being limited to a maximum of four tasks only. Another important limitation of this approach is that one has to decide the *order* in which the network trains on the tasks.

In this work we propose a fully on-line multi-task DRL approach that uses networks that are comparable in size to the single-task networks.

In particular, our contributions are the following: 1) We propose the first successful on-line multi-task learning framework which operates on MTIs that have many tasks with very visually different high-dimensional state spaces (See Figure 1 for a visual depiction of the 21 tasks that constitute our largest multi-tasking instance). 2) We present *three* concrete instantiations of our MTL framework: an adaptive method, a UCB-based meta-learning method and a A3C (Mnih et al., 2016a) based meta-learning method. 3) We propose a family of robust evaluation metrics for the multi-tasking problem and demonstrate that they evaluate a multi-tasking algorithm in a more sensible manner than existing metrics. 4) We provide extensive analyses of the abstract features learned by our methods and argue that most of the features help in generalization across tasks because they are task-agnostic. 5) We report results on *seven* distinct MTIs: three 6-task instances, one 8-task instance, two 12-task instances and one 21-task instance. Previous works have only reported results on a *single* MTI. Our largest MTI has more than *double* the number of tasks present in the largest MTI on which results have been published in the Deep RL literature (Rusu et al., 2016a). 6) We hence demonstrate how hyper-parameters tuned for an MTI (an instance with six tasks) generalize to other MTIs (with up to 21 tasks).

## 2  BACKGROUND

In this section, we introduce the various concepts needed to explain our proposed framework and the particular instantiations of the framework.

## 2.1 DISCOUNTED UCB1-TUNED+

A large class of decision problems can be cast as multi-armed bandit problem wherein the goal is to select the arm (or action) that gives the maximum expected reward. An efficient class of algorithms for solving the bandit problem are the UCB algorithms (Auer et al., 2002; Auer & Ortner, 2010). The UCB algorithms carefully track the uncertainty in estimates by giving exploration bonuses to the agent for exploring the lesser explored arms. Such UCB algorithms often maintain estimates for the average reward that an arm gives, the number of times the arm has been pulled and other exploration factors required to tune the exploration. In the case of non-stationary bandit problems, it is required for such average estimates to be non-stationary as well. For solving such non-stationary bandit problems, Discounted UCB style algorithms are often used (Kocsis & Szepesvári, 2006; Saito et al., 2014; Garivier & Moulines, 2011). In our UCB-based meta-learner experiments, we use the Discounted UCB1-Tuned+ algorithm (Kocsis & Szepesvári, 2006).

## 2.2 ACTOR CRITIC ALGORITHMS

One of the ways of learning optimal control using RL is by using (parametric) actor critic algorithms (Konda & Tsitsiklis, 2000). These approaches consist of two components: an actor and a critic. The actor is a parametric function: $\pi_{\theta_a}(a_t|s_t)$ mapping from the states to the actions according to which the RL agent acts ($\theta_a$ are the parameters of the policy/actor). A biased but low-variance sample estimate for the policy gradient is: $\nabla_{\theta_a} \log \pi_{\theta_a}(a_t|s_t)(Q(s_t, a_t) - b(s_t))$ where $a_t$ is the action executed in state $s_t$. $Q(s_t, a_t)$ is the action value function. $b(s_t)$ is a state-dependent baseline used for reducing the variance in policy gradient estimation. The critic estimates $Q(s_t, a_t)$ and possibly the state dependent baseline $b(s_t)$. Often, $b(s_t)$ is chosen to be the value function of the state, $V(s_t)$. We can also get an estimate for $Q(s_t, a_t)$ as $r_{t+1} + \gamma V(s_{t+1})$ where $\gamma$ is the discounting factor. The critic is trained using temporal difference learning (Sutton, 1988) algorithms like TD(0) (Sutton & Barto, 1998). The objective function for the critic is: $\mathbb{L}_{\theta_c} = (r_{t+1} + \gamma V(s_{t+1}) - V_{\theta_c}(s_t))^2$. In all our experiments, we use the Asynchronous Advantage Actor-Critic Algorithm (A3C) (Mnih et al., 2016a) as our base RL algorithm.

## 3 PROBLEM DEFINITION

In the domain of Multi-Task Learning, the goal is to obtain a single agent which can perform well on *all the k tasks* in a given fixed MTI. The performance metrics we use are presented in Section 3.1. While there might be transfer happening between the instances while learning, it is assumed that the agent at every point of time has access to all the tasks in the multi-task instance.

The MTA acts in an action space that is the union of the action spaces of the individual tasks. We assume that the input to the MTA is such that the state features are the same, or at the least the same feature learning mechanism will work across all the tasks. In this work, we demonstrate the effectiveness of our MTA on games from Arcade Learning Environment (Marc G. Bellemare et al., 2013). While these games are visually distinct, the same feature learning mechanism, namely a Convolutional Neural Network, works well across all the games.

It is also important to note that the identity of the current task is not part of the input to the MTN during training on a task. In contrast, existing methods such as (Rusu et al., 2016a) give the identity of the task that the MTN is being trained on as an input. Thus, our MTN must implicitly figure out the identity of the task just from the input features and the dynamics of the task.

## 3.1 EVALUATION METRICS

Previous works (Parisotto et al., 2016) define the performance of an MTA on an MTI as the arithmetic mean ($p_{am}$) of the normalized game-play scores of the MTA on the various tasks in the MTI. Let $\rho_i$ be the game-play score of an MTA in task $i$, $h_i$ be the target score in task $i$ (potentially obtained from other published work). We argue that $p_{am}$ is not a robust evaluation metric. An MTA can be as good as the target on all tasks and achieves $p_{am} = 1$. However, a bad MTA can achieve $p_{am} = 1$ by being $k$ (total number of tasks) times better than the target on **one** of the tasks and being as bad as getting 0 score in all the other tasks. We define a better performance metric: $q_{am}$ (Equation 1). It is better because the MTA needs to be good in *all* the tasks in order to get a high $q_{am}$. We also define, $q_{gm}$,

the geometric-mean based and $q_{hm}$, the harmonic-mean based performance metrics.

$$p_{am} = \left(\sum_{i=1}^{k} \frac{\rho_i}{h_i}\right) \Big/ k \qquad q_{am} = \left(\sum_{i=1}^{k} \min\left(\frac{\rho_i}{h_i}, 1\right)\right) \Big/ k \qquad q_{gm} = \sqrt[k]{\prod_{i=1}^{k} \min\left(\frac{\rho_i}{h_i}, 1\right)} \qquad (1)$$

$$q_{hm} = k \Big/ \left(\sum_{i=1}^{k} \max\left(\frac{h_i}{\rho_i}, 1\right)\right)$$

We evaluate the MTAs in our work on $p_{am}, q_{am}, q_{gm}, q_{hm}$. Table 1 reports the evaluation on $q_{am}$. Evaluations on the other metrics have been reported in Appendix $E$.

## 4 MODEL DEFINITION

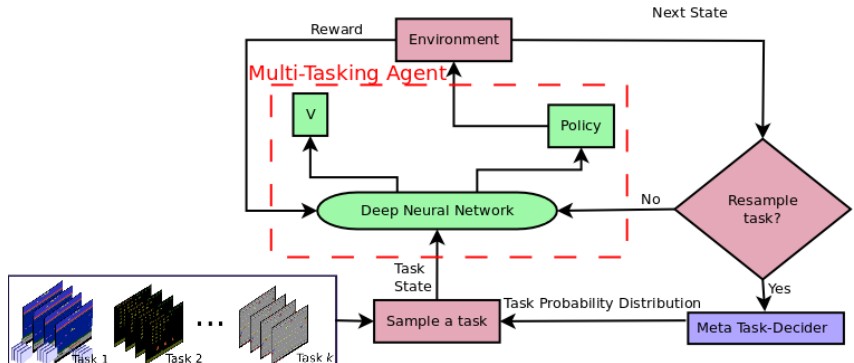

Figure 2: A visualization of our Active-sampling based Multi-Task Learning Framework

In this section, we introduce our framework for MTL by first describing a naive framework for on-line MTL in the first subsection and then presenting our approach as extension to this framework in the second subsection. To avoid the computational costs of training single-task expert networks, we assume that the MTA does not have access to expert networks' predictions. Previous approaches to MTL: (Parisotto et al., 2016; Rusu et al., 2016a) have been off-line in nature. Before we describe the frameworks, we outline how an *on-line* algorithm for MTL takes inputs from different tasks.

When a MTN is trained using an on-line MTL algorithm, it must be trained by interleaving data/observations from all the tasks in the MTI. An *on-line* MTL algorithm must decide once every few time steps, the next task on which the MTN is to be trained . We call such decision steps as *task decision steps*. Note that task decision steps can be event driven (eg: at the end of every episode) or time driven (eg: once every $k$ time steps).

### 4.1 BASELINE MULTI-TASKING AGENT

The BA3C is a simple on-line MTL algorithm. The MTN is a single A3C network which is trained by interleaving observations from $k$ tasks in an on-line fashion. The task decision steps in BA3C occur at the end of every episode of training for the MTN. The next task for the MTN to train on is decided uniformly at random. The full training algorithm for BA3C is given as Algorithm 2 in Appendix $C$. We believe that the lackluster performance of BA3C (this has been reported in (Parisotto et al., 2016) as well) is because of the *probability distribution* according to which the agent decides which task to train on next. In BA3C's case, this is the uniform probability distribution. We posit that this distribution is an important factor in determining the multi-tasking abilities of a trained MTA.

### 4.2 ACTIVE SAMPLING BASED MULTI-TASKING AGENT

We demonstrate our framework with the help of the LSTM (Hochreiter, Sepp et al., 1997) version of the A3C algorithm (Mnih et al., 2016a). Our framework is inspired by active learning principles

(Prince, 2004; Zhu, 2005; Settles, 2010). We call our framework **A4C- Active sampling A3C**. The overarching idea of our work is simple and effective: A multi-task learning algorithm can achieve better performance with fewer examples if it is *allowed to decide* which task to train on at every task decision step (thus "actively sampling" tasks) as opposed to uniformly sampling. More precisely, it is better if the agent decides to train on tasks which it is currently bad at. This decision can be made based on a heuristic or using another meta-learner. We explore two different approaches for the meta-learner - posing the meta-learning problem as an multi-arm bandit problem and as a full RL problem.

Figure 2 contains an illustration of our active-sampling based on-line MTL framework. Training algorithm for active sampling based on-line MTL is presented as Algorithm 1. Specific instantiations of Algorithm 1 for all of our proposed methods have been presented in Appendix $C$. The A4C MTN's

---

**Algorithm 1** Active Sampling based Multi-Task Learning

---

1: **function** MULTITASKING(SetOfTasks $T$)
2:     $h_i \leftarrow$ Target score in task $T_i$
3:     $n \leftarrow$ Number of episodes used for estimating current performance in any task $T_i$
4:     $s_i \leftarrow$ List of last $n$ scores that the multi-tasking agent scored during training on task $T_i$
5:     $p_i \leftarrow$ Probability of training on task $T_i$ next
6:     amta $\leftarrow$ The Active Sampling multi-tasking agent
7:     meta_decider $\leftarrow$ An instantiation of our active learning based task decision framework
8:     **for** train_steps:0 **to** MaxSteps **do**
9:         **for** $i$ in $\{1, \cdots, |T|\}$ **do**
10:            $\rho_i \leftarrow s_i$.average()
11:            $p_i \leftarrow$ meta_decider($\rho_i, h_i$)
12:         $j \sim p$   // Identity of the next task to train on
13:         $score_j \leftarrow$ amta.train($T_j$)
14:         $s_j$.append($score_j$)
15:         **if** $s_j$.length() $> n$ **then**
16:            $s_j$.remove_last()

---

architecture is the same as that of a single-task network. The important improvement is in the way the *next task for training* is selected. Instead of selecting the next task to train on uniformly at random, our framework maintains for each task $T_i$, an estimate of the MTN's current performance ($\rho_i$) as well as the target performance ($h_i$). These numbers are then used to actively sample tasks on which the MTA's current performance is poor. In all the methods we ensure that all tasks continue to be selected with non-zero probability during the learning. We emphasize that no single-task expert networks need to be trained for our framework; published scores from other papers (such as (Mnih et al., 2016a; Sharma et al., 2017) for Atari) or even Human performance can be used as target performance.

In case the task-decision problem is cast as a full RL problem, there are various definitions of state and reward that can be chosen. In what follows, we present 3 different **instantiations** of our A4C framework with particular choices of states and rewards. We experimented with other choices for state and reward definitions and we report the ones with the best performance in our experiments. There could be other agents under the A4C framework, some potentially better than our instantiations with other choices of state and reward functions and design of such agents are left as future work.

### 4.2.1 ADAPTIVE ACTIVE SAMPLING METHOD

We refer to this method as A5C (Adaptive Active-sampling A3C). The task decision steps in A5C occur at the end of every episode of training of the MTN. Among the methods we propose, this is the only method which does not learn the sampling distribution $p$ (Line 15, Algorithm 1). It computes an estimate of how well the MTN can solve task $T_i$ by calculating $m_i = \frac{h_i - \rho_i}{h_i}$ for each of the tasks. The probability distribution for sampling next tasks (at task decision steps) is then computed as: $p_i = \frac{e^{\frac{m_i}{\tau}}}{\sum_{c=1}^{k} e^{\frac{m_c}{\tau}}}$, where $\tau$ is a temperature hyper-parameter. Intuitively, $m_i$ is a task-invariant measure of how much the current performance of the MTN lags behind the target performance on task $T_i$. A higher value for $m_i$ means that the MTN is bad in Task $T_i$. By actively sampling from a

softmax probability distribution with $m_i$ as the *evidence*, our adaptive method is able to make smarter decisions about where to allocate the training resources next (which task to train on next).

### 4.2.2 UCB-BASED ACTIVE SAMPLING METHOD

This method is referred to as UA4C (UCB Active-sampling A3C). The task decision steps in UA4C occur at the end of every episode of training for the MTN. In this method, the problem of picking the next task to train on is cast as a multi-armed bandit problem with the different arms corresponding to the various tasks in the MTI being solved. The reward for the meta-learner is defined as: $r = m_i$ where $i$ is the index of the latest task that was picked by the meta-learner, and that the MTN was trained on. The reason for defining the reward in this way is that it allows the meta-learner to directly optimize for choosing those tasks that the MTN is bad at. For our experiments, we used the Discounted-UCB1-tuned+ algorithm (Kocsis & Szepesvári, 2006). We used a discounted-UCB algorithm because the bandit problem is non-stationary (the more a task is trained on, the smaller the rewards corresponding to the choice of that task become). We also introduced a tunable hyper-parameter $\beta$ which controls the relative importance of the bonus term and the average reward for a given task. Using the terminology from (Kocsis & Szepesvári, 2006), the upper confidence bound that the meta-learner uses for selecting the next task to train on is: $\bar{X}_t(\gamma, i) + \beta c_t(\gamma, i)$

### 4.2.3 EPISODIC META-LEARNER ACTIVE SAMPLING METHOD

We refer to this method as EA4C (Episodic meta-learner Active-sampling A3C). The task decision steps in EA4C occur at the end of every episode of training for the MTN (see Appendix $A$ for a version of EA4C which makes task-decision steps every few time steps of training). EA4C casts the problem of picking the next task to train on as a full RL problem. The idea behind casting it in this way is that by optimizing for the future sum of rewards (which are defined based on the multi-tasking performance) EA4C can learn the right *sequence* in which to sample tasks and hence learn a good *curriculum* for training MTN. The EA4C meta-learner consists of an LSTM-A3C-based controller that learns a task-sampling policy over the next tasks as a function of the previous sampling decisions and distributions that the meta-learner has used for decison making.

**Reward definition:** Reward for picking task $T_j$ at (meta-learner time step $t$) is defined as:

$$\text{rew}(T_j) = \lambda m_j + (1 - \lambda) \left( \frac{1}{3} \sum_{i \in \mathbb{L}} (1 - m_i) \right) \tag{2}$$

where $m_i$ was defined in Section 4.2.1. $\mathbb{L}$ is the set of worst three tasks, according to $1 - m_i = \frac{\rho_i}{h_i}$ (normalized task performance). $\lambda$ is a hyper-parameter. First part of the reward function is similar to that defined for the UCB meta-learner in Section 4.2.2. The second part of the reward function ensures that the performance of the MTN improves on worst three tasks and thus increases the multi-tasking performance in general.

**State Definition:** The state for the meta-learner is designed to be *descriptive* enough for it to learn the optimal policy over the choice of which task to train the MTN on next. To accomplish this, we pass a $3k$ length vector to the meta-learner (where $k$ is the number of tasks in the MTI) which is a concatenation of 3 vectors of length $k$. The first vector is a normalized count of the number of times each of the tasks has been sampled by the meta-learner since the beginning of training. The second vector is the identity of the task sampled at the previous *task decision step*, given as a one-hot vector. The third vector is the previous sampling distribution over the tasks that the meta learner had used to select the task on which the MTN was trained, at the last task decision step. Our definition of the meta-learner's state is just one of the many possible definitions. The first and the third vectors have been included to make the state descriptive. We included the identity of the latest task on which the MTN was trained so that the meta-learner is able to learn policies which are conditioned on the actual counts of the number of times a task was sampled.

## 5 EXPERIMENTAL SETUP AND RESULTS

The A3C MTNs we use have the same size and architecture as a single-task network (except when our MTL algorithms need to solve MT7, which has 21 tasks) and these architectural details are the same across different MTL algorithms that we present. The experimental details are meticulously

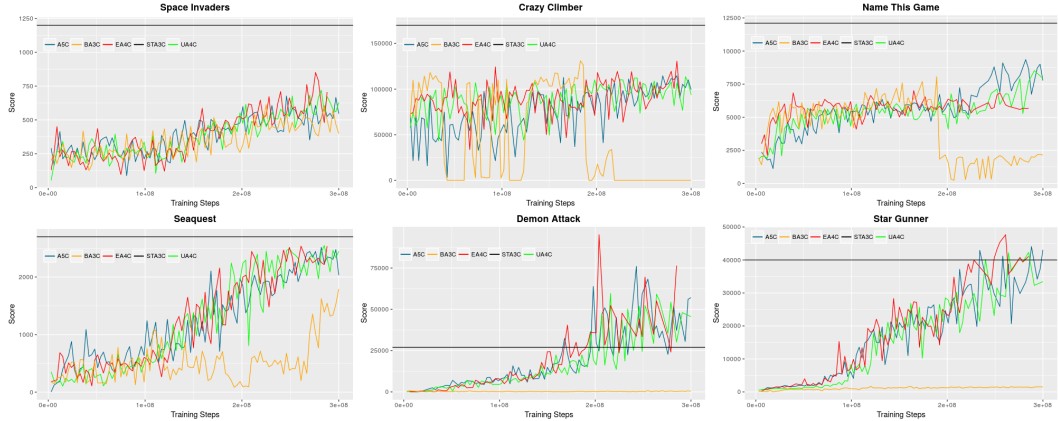

Figure 3: Evolution of game-play performance scores for A5C,EA4C,UA4C and BA3C agents. All training curves for all the multi-tasking instances have been presented in Appendix $D$.

documented in Appendix $B$. All our MTAs used a single 18-action shared output layer across all the different tasks in an MTI, instead of different output head agents per task as used in (Rusu et al., 2016a). Appendix $I$ contains our empirical argument against using such different-output head MTAs. It is important to note that previous works (Parisotto et al., 2016; Rusu et al., 2016a) have results only on a **single** MTI. Table 3 (in Appendix $B$) contains the description of the **seven** MTIs presented to MTAs in this work. Hyper-parameters for all multi-tasking algorithms in this work were tuned on only one MTI: MT1. If an MTI consists of $k$ tasks, then all MTNs in this work were trained on it for only $k \times 50$ million time steps, which is *half* of the combined training time for all the $k$ tasks put together (task-specific agents were trained for 100 million time steps in (Sharma et al., 2017)). All the target scores in this work were taken from Table 4 of (Sharma et al., 2017). We reiterate that for solving the MTL problem, it is not necessary to train single-task expert networks for arriving at the target scores; one can use scores published in other works.

We conducted experiments on *seven* different MTIs with number of constituent tasks varying from 6 to 21. All hyper-parameters were tuned on MT1, an MTI with 6 tasks. We demonstrate the robustness of our framework by testing all our the algorithms on seven MTIs including a 21-task MTI (MT7) which is **more than double** the number of tasks any previous work (Rusu et al., 2016a) has done multi-tasking on. Description of the MTIs used in this work is provided in Table 3 in Appendix $B$. We have performed three experiments to demonstrate the robustness of our method to the target scores chosen. These experiments and the supporting details have been documented in Appendix $G$. We now describe our findings from the general game-play experiments as demonstrated in Table 1. We observe that all our proposed models beat the BA3C agent by a large margin and obtain a performance of more than *double* the performance obtained by the BA3C agent. Among the proposed models, on MT1 (where the hyperparameters were tuned), A5C performs the best. However, the performance of UA4C and EA4C is only slightly lower than that of A5C. We accredit this relatively

Table 1: Comparison of the performance our MTAs to BA3C MTA based on $q_{am}$

|  | **MT1** | **MT2** | **MT3** | **MT4** | **MT5** | **MT6** | **MT7** |
|---|---|---|---|---|---|---|---|
| **\|T\|** | 6 | 6 | 6 | 8 | 12 | 12 | 21 |
| **A5C** | **0.799** | **0.601** | 0.646 | 0.915 | 0.650 | 0.741 | 0.607 |
| **UA4C** | 0.779 | 0.506 | **0.685** | 0.938 | **0.673** | **0.756** | 0.375 |
| **EA4C** | 0.779 | 0.585 | 0.591 | **0.963** | 0.587 | 0.730 | **0.648** |
| **BA3C** | 0.316 | 0.398 | 0.337 | 0.295 | 0.260 | 0.286 | 0.308 |

higher performance to the fact that there are many hyper-parameters to tune in the UA4C and the EA4C methods unlike A5C where only the temperature hyperparameter had to be tuned. We tuned all the important hyperparameters for UA4C and EA4C. However, our granularity of tuning was perhaps not very fine. This could be the reason for the slightly lower performance. The UA4C agent, however, generalizes better than A5C agent on the larger MTIs (MT5 & MT6). Also, the performance obtained by EA4C is close to that of A5C and UA4C in all the multitasking instances. The MTI MT4 has been taken from (Parisotto et al., 2016). On MT4, many of our agents are consistently able to obtain a performance close to $q_{am} = 0.9$. It is to be noted that Actor Mimic networks are only able to obtain $q_{am} = 0.79$ on the same MTI. The most important test of generalization is the 21-task instance (MT7). EA4C is by far the best performing method for this instance. This clearly demonstrates the hierarchy of generalization capabilities demonstrated by our proposed methods. At the first level, the EA4C MTA can learn task-agnostic representations which help it perform well on even large scale MTIs like MT7. Note that the hyper-parameters for all the algorithms were tuned on MT1, which is a 6-task instance. That the proposed methods can perform well on much larger instances with widely visually different constituent tasks without retuning hyperparameters is proof for a second level of generalization: the generalization of the hyper-parameter setting across multi-tasking instances.

## 6 ELIMINATING THE NEED FOR TARGET SCORES

An important component of our framework are the target scores for the different tasks. There are two concerns that one might have regarding the use of target scores: 1) Access to target scores implies access to trained single-task agents which defeats the purpose of *online* multi-task learning. 2) The method of training such an active-sampling based agent on new tasks where the tasks have *never* been solved. We aim to address both the concerns regarding the use of target scores in our proposed framework.

We reiterate that the access to target scores *does not* imply access to trained single-task agents. We would expect that any researcher who uses our framework would also use published resources as the source of target scores, rather than training single-task networks for each of the tasks.

In some cases, one might want to build an MTA *prior* to the existence of agents that can solve each of the single tasks. In such a case, it would be impossible to access target scores because the tasks in question have *never* been solved. In such cases, we propose to use a *doubling of targets* paradigm (demonstrated using Doubling UCB-based Active-sampling A3C (DUA4C) in Algorithm 7) to come up with rough estimates for the target scores and demonstrate that our doubling-target paradigm can result in impressive performances.

The doubling target paradigm maintains an *estimate* of target scores for each of the tasks that the MTA needs to solve. As soon as the MTA achieves a performance that is greater than or equal to the estimated target, the estimate for the target is doubled. The idea is that in some sense, the agent can keep improving until it hits a threshold, and then the threshold is doubled. All the hyper-parameters found by tuning UA4C on MT1 were retained. None of the hyper-parameters were retuned. This thus represents a setup which isn't very favorable for DUA4C.

Figure 4 depicts the evolution of the raw performance (game-score) of the DUA4C agent trained with doubling target estimates instead of single-task network's scores. The performance of DUA4C on different MTIs is contained in Table 2. Results on other metrics along with training curves on various MTIs are shown in Appendix K. We observe that even in this unfavorable setup, the performance of DUA4C is impressive. The performance could possibly improve if hyper-parameters were tuned for this specific paradigm/framework.

Table 2: Comparison of the performance DUA4C agent to BA3C based on $q_{am}$

|        | MT1   | MT2   | MT4   | MT5   |
|--------|-------|-------|-------|-------|
| **|T|**   | 6     | 6     | 8     | 12    |
| **DUA4C** | 0.661 | 0.533 | 0.576 | 0.509 |
| **BA3C**  | 0.316 | 0.398 | 0.295 | 0.260 |

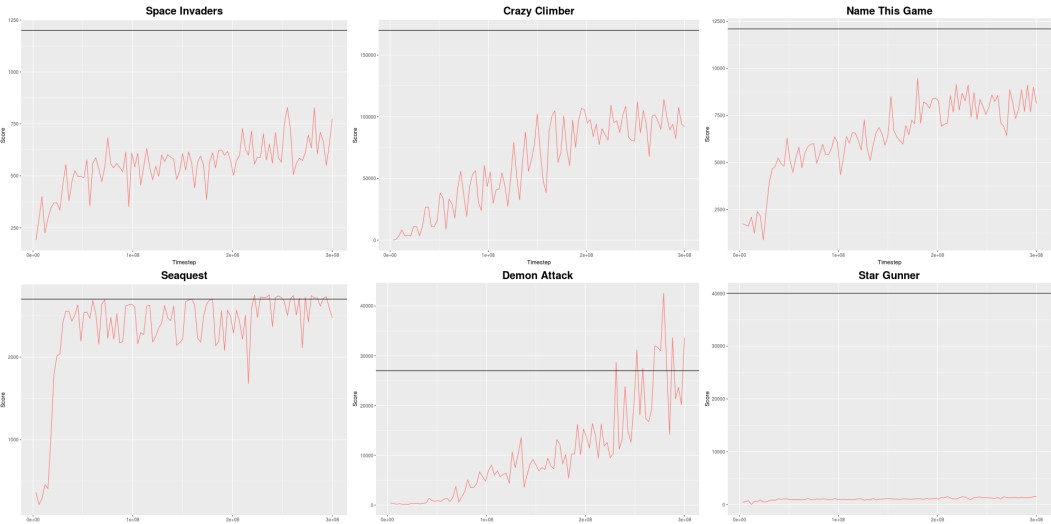

Figure 4: Training curve for DUA4C wherein doubling of target estimates is performed and thus no target scores whatsoever are used.

# 7 ANALYSIS OF TRAINED AGENTS

This section analyses the reasons as to why our MTL framework A4C performs much better than the baseline (BA3C). Based on the experiments that follow, we claim that it is the task-agnostic nature of the abstract features that are learned in this work which allow our proposed algorithms to perform very well. An MTA can potentially perform well at the different tasks in an MTI due to the *sheer representational power* of a deep neural network by learning *task-specific* features without *generalizing* across tasks. We empirically demonstrate that this is not the case for the agents proposed in our work. The experiments in this section analyze the activation patterns of the output of the LSTM controller. We call a neuron *task-agnostic* if it is as equally responsible for the performance on many of the tasks.

## 7.1 OVERCOMING CATASTROPHIC FORGETTING

Before we show the task agnostic nature of the neurons in our A4C agents, we present an intuition as to how our agents are able to overcome the problem of catastrophic forgetting. We first note that in all the agents defined under the A4C framework, a task has a higher probability to get sampled if the $m_i$ for the task is higher. Forgetting is avoided in our agents by the virtue of the sampling procedure used by the meta-learners. Say $m_1$ is largest among all $m_i$'s. This causes task 1 to get sampled more. Since the agent is training on task 1, it gets better at it. This leads to $m_1$ getting smaller. At some point if $m_2$(some other task) becomes larger than $m_1$, task 2 will start getting sampled more. At some later point, if performance on task 1 degrades due to the changes made to the network, then $m_1$ will again become larger and thus it'll start getting sampled more. It can now be argued that performance estimates ($m_i$) could be stale for some tasks if they don't get sampled. While it is true that we don't update the score of a task till it is sampled again, we need to keep in mind that the sampling of the tasks is done from a distribution across tasks. As a result, there is still finite probability of every task getting sampled. This is analogous to exploration in RL. Note that if the probability of sampling such tasks was so low that it would practically be impossible to sample it again, this would imply that performance on the task was great. What we have observed through comprehensive experimentation is that once such good performance has been achieved on some task, degradation does not happen.

## 7.2 ACTIVATION DISTRIBUTION OF NEURONS

In this set of experiments, our agents trained on $MT_1$ are executed on each of the constituent tasks for 10 episodes. A neuron is said to fire for a time step if its output has an absolute value of $0.3$ or more.

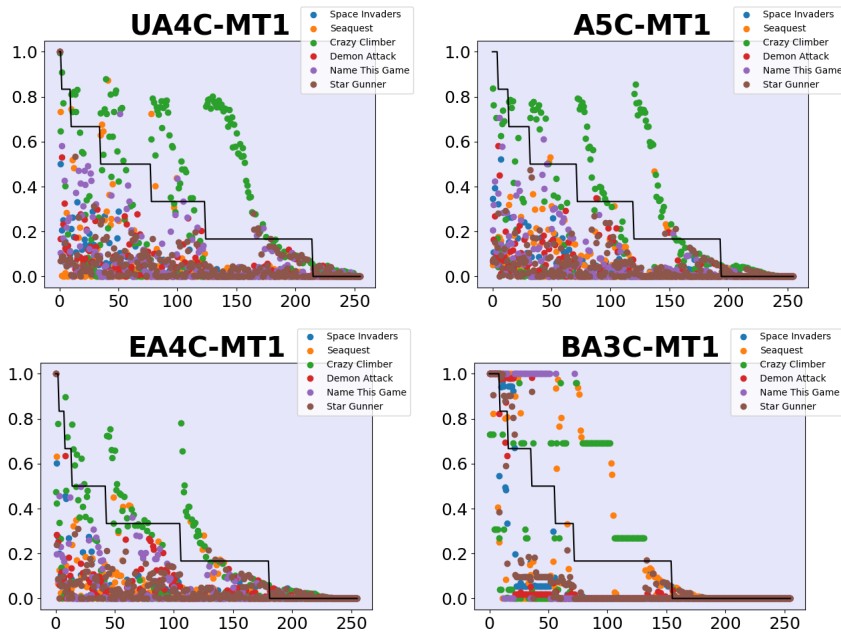

Figure 5: Understanding abstract LSTM features in our proposed methods by analyzing firing patterns

Let $f_{ij}$ denote the fraction of time steps for which neuron $j$ fires when tested on task $i$. Neuron $j$ fires for the task $i$ if $f_{ij} \geq 0.01$. We chose this low threshold because there could be important neurons that detect rare events. Figure 5 demonstrates that for A4C, a large fraction of the neurons fire for a large subset of tasks and are not task-specific. It plots neuron index versus the set of fraction of time steps that neuron fires in, for each task. The neurons have been sorted first by $|\{i : f_{ij} \geq 0.01\}|$ and then by $\sum_i f_{ij}$. Neurons to the left of the figure fire for many tasks whereas those towards the right are task-specific. The piece-wise constant line in the figure counts the number of tasks in which a particular neuron fires with the leftmost part signifying 6 tasks and the rightmost part signifying zero tasks. Appendix $H$ contains the analysis for all MTIs and methods.

### 7.3 TURNOFF ANALYSIS

We introduce a way to analyze multitasking agents without using any thresholds. We call this method the *turnoff-analysis*. Here, we force the activations of one of the neurons in LSTM output to $0$ and then observe the change in the performances on individual tasks with the neuron switched off. This new score is then compared with the original score of the agent when none of the neurons were switched off and an absolute percentage change in the scores is computed. These percentage changes are then normalized for each neuron and thus a tasks versus neuron matrix $A$ is obtained. The variance of column $i$ of $A$ gives a score for the task-specificity of the neuron $i$. We then sort the columns of $A$ in the increasing order of variance and plot a heatmap of the matrix $A$. We conclude from Figure 6 that A4C agents learn many non task-specific abstract features which help them perform well across a large range of tasks. Our experiments demonstrate that A4C agents learn many more task-agnostic abstract features than the BA3C agent. Specifically, observe how uniformly pink the plot corresponding to the UA4C agent is, compared to the BA3C plot.

## 8 CONCLUSIONS AND FUTURE WORK

We propose a framework for training MTNs which , through a form of active learning succeeds in learning to perform on-line multi-task learning. The key insight in our work is that by choosing the task to train on, an MTA can choose to concentrate its resources on tasks in which it currently performs poorly. While we do not claim that our method solves the problem of on-line multi-task reinforcement learning definitively, we believe it is an important first step. Our method is complementary to many

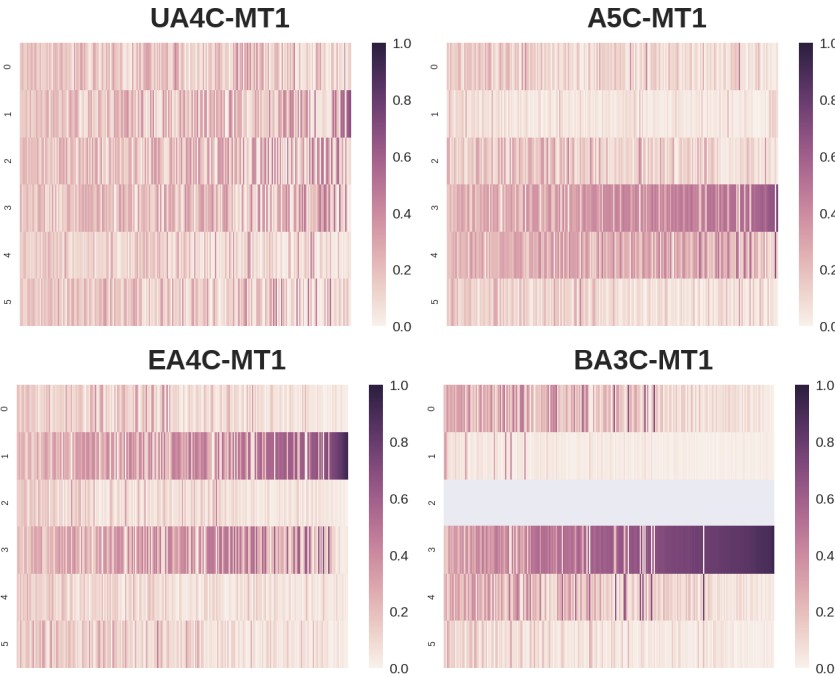

Figure 6: Turn Off analysis heap-maps for the all agents. For BA3C since the agent scored 0 on one of the games, normalization along the neuron was done only across the other 5 games.

of the existing works in the field of multi-task learning such as: (Rusu et al., 2016a) and (Parisotto et al., 2016). These methods could potentially benefit from our work. Another possible direction for future work could be to explicitly force the learned abstract representations to be task-agnostic by imposing objective function based regularizations. One possible regularization could be to force the average firing rate of a neuron to be the same across the different tasks.

## ACKNOWLEDGMENTS

The authors would like to thank Dr. Mitesh Khapra for many comments and suggestions on the paper. We would also like to thank the anonymous reviewers for their constructive feedback. We would also like to thank the Amazon Web Services(AWS) Educate program for providing us with the computational resources for the experiment. This work is supported by a funding from Robert Bosch Centre for Data Science and Artificial Intelligence (RBC-DSAI) at IIT Madras. The second first author's travel to ICLR is supported by a grant from Google.

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

APPENDIX A: FINE-GRAINED META-LEARNER ACTIVE SAMPLING METHOD

In the EA4C method introduced in Section 4.2.3, the task-decision steps, which also correspond to one training step for the meta-learner, happen at the end of one episode of training on one of the tasks. For three of the multi-tasking instances (MT1, MT2 and MT3) that we experimented with, the total number of training steps was 300 million. Also, an average episode length of tasks in these instances is of the order of 1000 steps. Hence, the number of training steps for the meta-learner in EA4C is of the order of $3 \times 10^5$. This severely restricts the size of the neural network which is used to represent the policy of the meta-learner.

To alleviate this problem we introduce a method called FA4C: Fine-grained meta-learner Active-sampling A3C. The same architecture and training procedure from EA4C is used for FA4C, except for the fact that task decision steps happen after every $N$ steps of training the multi-tasking network, instead of at the end of an episode. The value of $N$ was fixed to be 20. This is the same as the value of $n$ used for $n$-step returns in our work as well as (Sharma et al., 2017). Observe that when the number of training steps for the multi-tasking network is 300 million, the number of training steps for meta-learner is now of the order of 15 million. This allows the use of larger neural networks for the meta-learner policy as compared to EA4C. Since we used an LSTM in the neural network representing the multitasking agent's policy, we stored the state of the LSTM cells at the end of these $n = 20$ training steps for each of the tasks. This allows us to resume executing any of the tasks after training on one of them for just 20 steps using these cached LSTM state cells.

We now describe the reward function and state construction for FA4C:

**Reward Function**: Since the task decision steps for this method happen after every 20 steps of training the multi-tasking network, the meta-learner needs to be rewarded in a way that evaluates its 20-step task selection policy. It makes sense to define this reward to be proportional to the performance of the MTN during **those** 20 time steps, and inversely proportional to the target performance during those 20 time steps. These target scores have to be computed differently from those used by other methods introduced in this paper since the scores now correspond to performance over twenty time steps and not over the entire episode.

The target scores for a task in FA4C can be obtained by summing the score of a trained single-task agent over twenty time steps and finding the value of this score averaged over the length of the episode. Concretely, if the single-task agent is executed for $k$ episodes and each episode is of length $l_i, \ 1 \le i \le k$ and $r_{i,j}$ denotes the reward obtained by the agent at time step $j$ in episode $i$ where $1 \le i \le k, \ 1 \le j \le l_i$ then the averaged 20-step target score is given by (let $x_i = \lfloor \frac{l_i}{20} \rfloor$):

$$h_{fg} = \frac{\sum\limits_{i=1}^{k} \dfrac{\sum\limits_{j=0}^{x_i-1} \sum\limits_{m=1}^{20} r_{i,20j+m}}{x_i}}{k} \tag{3}$$

This design of the target score has two potential flaws:

1) A task could be very rewarding in certain parts of the state space(and hence during a particular period of an episode) whereas it could be inherently sparsely rewarding over other parts. It would thus make sense to use different target scores for different parts of the episode. However, we believe that in an expected sense our design of the target score is feasible.

2) Access to such fine grained target scores is hard to get. While the target scores used in the rest of the paper are simple scalars that we took from other published work (Sharma et al., 2017), for getting these $h_{fg}$'s we had to train single-task networks and get these fine grained target scores. Hopefully such re-training for targets would not be necessary once a larger fraction of the research starts open-sourcing not only their codes but also their trained models.

The overall reward function is the same as that for EA4C (defined in Equation 2) except one change, $m_i$ is now defined as:

$$m_{i,fg} = \frac{\rho_{i,fg}}{h_{i,fg}}$$

where $h_{i,fg}$ is the target score defined in Equation 3 for task $T_i$ and $\rho_{i,fg}$ is the score obtained by multi-tasking instance in task $T_i$ over a duration of twenty time steps.

**State Function**: The state used for the fine-grained meta-learner is the same as that used by the episodic meta-learner.

Our experimental results show that while FA4C is able to perform better than random on some multi-tasking instances, on others, it doesn't perform very well. This necessitates a need for better experimentation and design of fine-grained meta controllers for multi-task learning.

## APPENDIX B: EXPERIMENTAL DETAILS

We first describe the seven multi-tasking instances with which we experiment. We then describe the hyper-parameters of the MTA which is common across all the 5 methods (BA3C, A5C, UA4C, EA4C, FA4C) that we have experimented with, in this paper. In the subsequent subsections we describe the hyper-parameter choices for A5C, UA4C, EA4C, FA4C.

### MULTI-TASKING INSTANCES

The seven multi-tasking instances we experimented with have been documented in Table 3. The first three instances are six task instances, meant to be the smallest instances. MT4 is an 8-task instance. It has been taken from (Parisotto et al., 2016) and depicts the 8 tasks on which (Parisotto et al., 2016) experimented. We experimented with this instance to ensure that some kind of a comparison can be carried out on a set of tasks on which other results have been reported. MT5 and MT6 are 12-task instances and demonstrate the generalization capabilities of our methods to medium-sized multi-tasking instances. Note that even these multi-tasking instances have two more tasks than any other multi-tasking result (Policy distillation (Rusu et al., 2016a) reports results on a 10-task instance. However, we decided not to experiment with that set of tasks because the result has been demonstrated with the help of a neural network which is 4 times the size of a single task network. In comparison, all of our results for 6, 8 and 12 task instances use a network which has same size as a single-task network). Our last set of experiments are on a 21-task instance. This is in some sense a holy grail of multi-tasking since it consists of 21 extremely visually different tasks. The network used for this set of experiments is only twice the size of a single-task network. Hence, the MTA still needs to distill the prowess of 10.5 tasks into the number of parameters used for modeling a single-task network. Note that this multi-tasking instance is more than twice the size of any other previously published work in multi-tasking.

Table 3: Descriptions of the seven multi-tasking instances.

|  | MT1 | MT2 | MT3 | MT4 | MT5 | MT6 | MT7 |
|---|---|---|---|---|---|---|---|
| T1 | Space Invaders | Asterix | Breakout | Atlantis | Space Invaders | Atlantis | Space Invaders |
| T2 | Seaquest | Alien | Centipede | Breakout | Seaquest | Amidar | Seaquest |
| T3 | Crazy Climber | Assault | Frostbite | Bowling | Asterix | Breakout | Asterix |
| T4 | Demon Attack | Time Pilot | Q-bert | Crazy Climber | Alien | Bowling | Alien |
| T5 | Name This Game | Gopher | Kung Fu Master | Seaquest | Assault | Beam Rider | Assault |
| T6 | Star Gunner | Chopper Command | Wizard of Wor | Space Invaders | Bank Heist | Chopper Command | Bank Heist |
| T7 |  |  |  | Pong | Crazy Climber | Centipede | Crazy Climber |
| T8 |  |  |  | Enduro | Demon Attack | Frostbite | Demon Attack |
| T9 |  |  |  |  | Gopher | Kung Fu Master | Gopher |
| T10 |  |  |  |  | Name This game | Pong | Name This game |
| T11 |  |  |  |  | Star Gunner | Road Runner | Star Gunner |
| T12 |  |  |  |  | TutanKham | Phoenix | TutanKham |
| T13 |  |  |  |  |  |  | Amidar |
| T14 |  |  |  |  |  |  | Chopper Command |
| T15 |  |  |  |  |  |  | Breakout |
| T16 |  |  |  |  |  |  | Beam Rider |
| T17 |  |  |  |  |  |  | Bowling |
| T18 |  |  |  |  |  |  | Centipede |
| T19 |  |  |  |  |  |  | Krull |
| T20 |  |  |  |  |  |  | Kangaroo |
| T21 |  |  |  |  |  |  | Phoenix |

MULTI-TASKING AGENT

In this sub-section we document the experimental details regarding the MTN that we used in our experiments.

SIMPLE HYPER-PARAMETERS

We used the LSTM version of the network proposed in (Mnih et al., 2016a) and trained it using the async-rms-prop algorithm. The initial learning rate was set to $10^{-3}$ (found after hyper-parameter tuning over the set $\{7 \times 10^{-4}, 10^{-3}\}$) and it was decayed linearly over the entire training period to a value of $10^{-4}$. The value of $n$ in the $n$-step returns used by A3C was set to 20. This found after hyper-parameter tuning over the set $\{5, 20\}$. The discount factor $\gamma$ for the discounted returns was set to be $\gamma = 0.99$. Entropy-regularization was used to encourage exploration, similar to its use in (Mnih et al., 2016a). The hyper-parameter which trades-off optimizing for the entropy and the policy improvement is $\beta$ (introduced in (Mnih et al., 2016a). $\beta = 0.02$ was separately found to give the best performance for all the active sampling methods (A5C, UA4C, EA4C, FA4C) after hyper-parameter tuning over the set $\{0.003, 0.01, 0.02, 0.03, 0.05\}$. The best $\beta$ for BA3C was found to be $0.01$.

The six task instances (MT1, MT2 and MT3) were trained for 300 million steps. The eight task instance (MT4) was trained over 400 million steps. The twelve task instances (MT5 and MT6) were trained for 600 million steps. The twenty-one task instance was trained for 1.05 billion steps. Note that these training times were chosen to ensure that each of our methods was at least $50\%$ more data efficient than competing methods such as off-line policy distillation. All the models on all the instances ,except the twenty-one task instance were trained with 16 parallel threads. The models on the twenty-one task instance were trained with 20 parallel threads.

Training and evaluation were interleaved. It is to be noted that while during the training period active sampling principles were used in this work to improve multi-tasking performance, during the testing/evaluation period, the multi-tasking network executed on each task for the same duration of time (5 episodes, each episode capped at length 30000). For the smaller multi-tasking instances (MT1, MT2, MT3 and MT4), after every 3 million training steps, the multi-tasking network was made to execute on each of the constituent tasks of the multi-tasking instance it is solving for 5 episodes each. Each such episode was capped at a length of 30000 to ensure that the overall evaluation time was bounded above. For the larger multi-tasking instances (MT5, MT6 and MT7) the exact same procedure was carried out for evaluation, except that evaluation was done after every 5 million training steps. The lower level details of the evaluation scheme used are the same as those described in (Sharma et al., 2017). The evolution of this average game-play performance with training progress has been demonstrated for MT1 in Figure 3. Training curves for other multi-tasking instances are presented in Appendix $D$.

ARCHITECTURE DETAILS

We used a low level architecture similar to (Mnih et al., 2016a; Sharma et al., 2017) which in turn uses the same low level architecture as (Mnih et al., 2015). The first three layers of are convolutional layers with same filter sizes, strides, padding as (Mnih et al., 2015; 2016a; Sharma et al., 2017). The convolutional layers each have 64 filters. These convolutional layers are followed by two fully connected (FC) layers and an LSTM layer. A policy and a value function are derived from the LSTM outputs using two different output heads. The number of neurons in each of the FC layers and the LSTM layers is 256.

Similar to (Mnih et al., 2016a) the Actor and Critic share all but the final layer. Each of the two functions: policy and value function are realized with a different final output layer, with the value function outputs having no non-linearity and with the policy having a softmax-non linearity as output non-linearity, to model the multinomial distribution.

We will now describe the hyper-parameters of the *meta-task-decider* used in each of the methods proposed in the paper:

## A5C

### SIMPLE HYPER-PARAMETERS

The algorithm for A5C has been specified in Algorithm 3. The temperature parameter $\tau$ in the softmax function used for the task selection was tuned over the set $\{0.025, 0.033, 0.05, 0.1\}$. The best value was found to be $0.05$. Hyper-parameter $n$ was set to be $10$. The hyper-parameter $l$ was set to be $4$ million.

## UA4C

### SIMPLE HYPER-PARAMETERS

The discounted UCB1-tuned + algorithm from (Kocsis & Szepesvári, 2006) was used to implement the *meta-task-decider*. The algorithm for training UA4C agents has been demonstrated in Algorithm 4. We hyper-parameter tuned for the discount factor $\gamma$ used for the meta-decider (tuned over the set $\{0.8, 0.9, 0.99\}$) and the scaling factor for the bonus $\beta$ (tuned over the set $\{0.125, 0.25, 0.5, 1\}$). The best hyper-parameters were found to be $\gamma = 0.99$ and $\beta = 0.25$.

## EA4C

### SIMPLE HYPER-PARAMETERS

The meta-learner network was also a type of A3C network, with one meta-learner thread being associated with one multi-task learner thread. The task that the MTN on thread $i$ trained on was sampled according to the policy of the meta-learner $M_i$ where $M_i$ denotes the meta-learner which is executing on thread $i$. The meta-learner was also trained using the A3C algorithm with async-rms-prop. The meta-learner used 1-step returns instead of the 20-step returns that the usual A3C algorithm uses. The algorithm for training EA4C agents has been demonstrated in Algorithm 5. We tuned the $\beta_{\text{meta}}$ for entropy regularization for encouraging exploration in the meta-learner's policy over the set $\{0, 0.003, 0.01\}$ and found the best value to be $\beta_{\text{meta}} = 0$. We also experimented with the $\gamma_{\text{meta}}$, the discounting factor for the RL problem that the meta-learner is solving. We tuned it over the set $\{0.5, 0.8, 0.9\}$ and found the best value to be $\gamma_{\text{meta}} = 0.8$. The initial learning rate for the meta learner was tuned over the set $\{5 \times 10^{-4}, 10^{-3}, 3 \times 10^{-3}\}$ and $10^{-3}$ was found to be the optimal initial learning rate. Similar to the multi-tasking network, the learning rate was linearly annealed to $10^{-4}$ over the number of training steps for which the multi-tasking network was trained.

ARCHITECTURAL DETAILS

We extensively experimented with the architecture of the meta-learner. We experimented with feed-forward and LSTM versions of EA4C and found that the LSTM versions comprehensively outperform the feed-forward versions in terms of the multi-tasking performance ($q_{am}$). We also comprehensively experimented with wide, narrow, deep and shallow networks. We found that increasing depth beyond a point ($\geq 3$ fully connected layers) hurt the multi-tasking performance. Wide neural networks (both shallow and deep ones) were unable to perform as well as their narrower counter-parts. The number of neurons in a layer was tuned over the set $\{50, 100, 200, 300, 500\}$ and $100$ was found to be the optimal number of neurons in a layer. The number of fully-connected layers in the meta-learner was tuned over the set $\{1, 2, 3\}$ was 2 was found to be the optimal depth of the meta-controller. The best-performing architecture of the meta-learner network consists of: two fully-connected layers with 100 neurons each, followed by an LSTM layer with 100 LSTM cells, followed by one linear layer each modeling the meta-learner's policy and its value function. We experimented with dropout layers in meta-learner architecture but found no improvement and hence did not include it in the final architecture using which all experiments were performed.

FA4C

All the hyper-parameters for FA4C were tuned in exactly the same way that they were tuned for EA4C. The *task decision steps* for FA4C were time-driven (taken at regular intervals of training the multi-tasking network) rather than being event-driven (happening at the end of an episode, like in the EA4C case). While the interval corresponding to the task decision steps can in general be different from the $n$ to be used for $n$-step returns, we chose both of them to be the same with $n = 20$. This was done to allow for an easier implementation of the FA4C method. Also, 20 was large enough so that one could find meaningful estimates of 20-step cumulative returns without the estimate having a high variance and also small enough so that the FA4C meta-learner was allowed to make a large number of updates (when the multi-tasking networks were trained for 300 million steps (like in MT1, MT2 and MT3) The FA4C meta-learner was trained for roughly 15 million steps.)

APPENDIX C: TRAINING ALGORITHMS FOR OUR PROPOSED METHODS

Algorithm 1 contains a pseudo-code for training a generic active sampling method proposed in this work. This appendix contains specific instantiations of that algorithm for the all the methods proposed in this work. It also contains an algorithm for training the baseline MTA proposed in this work.

TRAINING BASELINE MULTI-TASKING NETWORKS (BA3C)

---

**Algorithm 2** Baseline Multi-Task Learning

---

1: **function** BASELINEMULTITASKING( SetOfTasks T )
2:     $k \leftarrow |T|$
3:     $t \leftarrow$ Total number of training steps for the algorithm
4:     bmta $\leftarrow$ the baseline multi-tasking agent
5:     **for** $i$ in $\{1, \cdots, k\}$ **do**
6:         $p_i \leftarrow \frac{1}{k}$
7:     **for** train_steps:0 **to** $t$ **do**
8:         $j \leftarrow j \sim p_i$. Identity of next task to train on
9:         $score_j \leftarrow$ bsmta.train_for_one_episode($T_j$)

---

TRAINING ADAPTIVE ACTIVE-SAMPLING AGENTS (A5C)

---

**Algorithm 3** A5C

---

1: **function** MULTITASKING( SetOfTasks T )
2:     $k \leftarrow |T|$
3:     $h_i \leftarrow$ Target score in task $T_i$. This could be based on expert human performance or even published scores from other technical works
4:     $n \leftarrow$ Number of episodes which are used for estimating current average performance in any task $T_i$
5:     $l \leftarrow$ Number of training steps for which a uniformly random policy is executed for task selection. At the end of $l$ training steps, the agent must have learned on $\geq n$ episodes $\forall$ tasks $T_i \in T$
6:     $t \leftarrow$ Total number of training steps for the algorithm
7:     $s_i \leftarrow$ List of last $n$ scores that the multi-tasking agent scored during training on task $T_i$.
8:     $p_i \leftarrow$ Probability of training on an episode of task $T_i$ next.
9:     $\tau \leftarrow$ Temperature hyper-parameter of the softmax task-selection non-parametric policy
10:    amta $\leftarrow$ The Active Sampling multi-tasking agent
11:    **for** $i$ in $\{1, \cdots, k\}$ **do**
12:        $p_i \leftarrow \frac{1}{k}$
13:    **for** train_steps:0 **to** $t$ **do**
14:        **if** train_steps $\geq l$ **then**
15:            **for** $i$ in $\{1, \cdots, k\}$ **do**
16:                $\rho_i \leftarrow s_i$.average()
17:                $m_i \leftarrow \frac{h_i - \rho_i}{h_i \times \tau}$
18:                $p_i \leftarrow \frac{e^{m_i}}{\sum_{q=1}^{k} e^{m_q}}$
19:        $j \sim p$     // The next task on which the multi-tasking agent is trained
20:        $score_j \leftarrow$ amta.train_for_one_episode($T_j$)
21:        $s_j$.append($score_j$)
22:        **if** $s_j$.length() $> n$ **then**
23:            $s_j$.remove_oldest()     // Removes the oldest score for task $T_j$

---

TRAINING UCB-BASED META-LEARNER (UA4C)

---

**Algorithm 4** UA4C

---

1: **function** MULTITASKING(SetOfTasks T)
2:     $k \leftarrow |T|$
3:     $h_i \leftarrow$ Target score in task $T_i$. This could be based on expert human performance or even published scores from other technical works
4:     $t \leftarrow$ Total number of training steps for the algorithm
5:     $\gamma \leftarrow$ Discount factor for Discounted UCB
6:     $\beta \leftarrow$ Scaling factor for exploration bonuses
7:     $X_i \leftarrow$ Discounted sum of rewards for task $i$
8:     $\bar{X}_i \leftarrow$ Mean of discounted sum of rewards for task $i$
9:     $c_i \leftarrow$ Exploration bonus for task $i$
10:    $n_i \leftarrow$ Discounted count of the number of episodes the agent has trained on task $i$
11:    amta $\leftarrow$ The Active Sampling multi-tasking agent
12:    $X_i \leftarrow 0 \ \forall i$
13:    $c_i \leftarrow 0 \ \forall i$
14:    $n_i \leftarrow 0 \ \forall i$
15:    $\bar{X}_i \leftarrow 0 \ \forall i$
16:    **for** train_steps:0 **to** $t$ **do**
17:        $j \leftarrow \mathrm{argmax}_l \ \bar{X}_l + \beta c_l$    // The next task on which the multi-tasking agent is trained
18:        score $\leftarrow$ amta.train_for_one_episode($T_j$)
19:        $X_i \leftarrow \gamma X_i \ \forall i$
20:        $X_j \leftarrow X_j + \max\left(\frac{h_j - score}{h_j}, 0\right)$
21:        $n_i \leftarrow \gamma n_i \ \forall i$
22:        $n_j \leftarrow n_j + 1$
23:        $\bar{X}_i \leftarrow X_i/n_i \ \forall i$
24:        $c_i \leftarrow \sqrt{\frac{\max(\bar{X}_i(1-\bar{X}_i), 0.002) \times \log(\sum_l n_l)}{n_i}} \ \forall i$

---

TRAINING EPISODIC META-LEARNER BASED ACTIVE SAMPLING AGENT (EA4C)

---

**Algorithm 5** EA4C

---

1: **function** MULTITASKING( SetOfTasks T )
2:     $k \leftarrow |T|$
3:     $h_i \leftarrow$ Target score in task $T_i$. This could be based on expert human performance or even published scores from other technical works
4:     $n \leftarrow$ Number of episodes which are used for estimating current average performance in any task $T_i$
5:     $t \leftarrow$ Total number of training steps for the algorithm
6:     $s_i \leftarrow$ List of last $n$ scores that the multi-tasking agent scored during training on task $T_i$.
7:     $p_i \leftarrow$ Probability of training on an episode of task $T_i$ next.
8:     $c_i \leftarrow$ Count of the number of training episodes of task $T_i$
9:     $r_1, r_2 \leftarrow$ First & second component of the reward for meta-learner, defined in lines 24-25
10:     $r \leftarrow$ Reward for meta-learner
11:     $l \leftarrow$ Number of tasks to consider in the computation of $r_2$
12:     amta $\leftarrow$ The Active Sampling multi-tasking agent
13:     ma $\leftarrow$ Meta Learning Agent
14:     $p_i \leftarrow \frac{1}{k} \ \forall i$    // First task for meta learner training is decided uniformly at random
15:     **for** train_steps:0 **to** $t$ **do**
16:         $j \sim p$    // Identity of the next task on which the multi-tasking agent is trained
17:         $c_j \leftarrow c_j + 1$
18:         $score_j \leftarrow$ amta.train_for_one_episode$(T_j)$
19:         $s_j$.append$(score_j)$
20:         **if** $s_j$.length$() > n$ **then**
21:             $s_j$.remove_oldest()    // Removes the oldest score for task $T_j$
22:         $s_{\text{avg}_i} \leftarrow$ average$(s_i) \ \forall i$
23:         $s_{\text{min-l}} \leftarrow$ sort_ascending$(s_{\text{avg}_i}/h_i)[0:l]$    // Find $l$ smallest normalized scores
24:         $r_1 \leftarrow 1 - s_j/b_j$
25:         $r_2 \leftarrow 1 -$ average$(\text{clip}(s_{\text{min-l}}, 0, 1))$
26:         $r \leftarrow \lambda r_1 + (1 - \lambda)r_2$
27:         $p \leftarrow$ mt.train_and_sample$\left(\text{state} = \left[\frac{c}{\text{sum}(c)}, p, \text{one\_hot}(j)\right], \text{reward} = r\right)$

---

TRAINING FINE-GRAINED META-LEARNER BASED ACTIVE SAMPLING AGENT (FA4C)

---

**Algorithm 6** FA4C

---

1: **function** MULTITASKING( SetOfTasks T )
2:     $k \leftarrow |T|$
3:     $N \leftarrow$ Length of fine-grained episode
4:     $h_i^{fg} \leftarrow$ Fine-grained Target score in task $T_i$. This could be based on expert human
         performance
5:     $n \leftarrow$ Number of episodes which are used for estimating current average performance in
         any task $T_i$
6:     $t \leftarrow$ Total number of training steps for the algorithm
7:     $s_i \leftarrow$ List of last $n$ scores that the multi-tasking agent scored during training on task $T_i$.
8:     $p_i \leftarrow$ Probability of training on an episode of task $T_i$ next.
9:     $c_i \leftarrow$ Count of the number of training episodes of task $i$
10:    $r_1, r_2 \leftarrow$ First & second component of the reward for meta-learner, defined in lines 24-25
11:    $r \leftarrow$ Reward for meta-learner
12:    $l \leftarrow$ Number of tasks to consider in the computation of $r_2$
13:    ma $\leftarrow$ Meta Learning Agent
14:    amta $\leftarrow$ The Active Sampling multi-tasking agent
15:    $p_i \leftarrow \frac{1}{k} \ \forall i$    // First task for meta learner training is decided uniformly at random
16:    **for** train_steps:0 **to** $t$ **do**
17:       $j \sim p$    // Identity of the next task on which the multi-tasking agent is trained
18:       $c_j \leftarrow c_j + 1$
19:       $\text{score}_j \leftarrow \text{amta.train}(T_j, \text{steps} = N)$    // Save the LSTM state for task $T_j$ until the
                                                    // next time this task is resumed
20:       $s_j.\text{append}(\text{score}_j)$
21:       **if** $s_j.\text{length}() > n$ **then**
22:          $s_j.\text{remove\_oldest}()$    // Removes the oldest score for task $T_j$
23:       $s_{\text{avg}_i} \leftarrow \text{average}(s_i) \ \forall i$
24:       $s_{\text{min-l}} \leftarrow \text{sort\_ascending}(s_{\text{avg}_i}/h_i^{fg})[0:l]$    // Find $l$ smallest normalized scores
25:       $r_1 \leftarrow 1 - s_j/h_j^{fg}$
26:       $r_2 \leftarrow 1 - \text{average}(\text{clip}(s_{\text{min-l}}, 0, 1))$
27:       $r \leftarrow \lambda r_1 + (1 - \lambda)r_2$
28:       $p \leftarrow \text{ma.train\_and\_sample}\left(\text{state} = \left[\frac{c}{\text{sum}(c)}, p, \text{one\_hot}(j)\right], \text{reward} = r\right)$

---

TRAINING DOUBLING UCB BASED ACTIVE SAMPLING AGENT (DUA4C)

---

**Algorithm 7** DUA4C

---
1: **function** MULTITASKING( SetOfTasks T )
2:     $k \leftarrow |T|$
3:     $h_i \leftarrow$ Estimated target for task $T_i$
4:     $t \leftarrow$ Total number of training steps for the algorithm
5:     $\gamma \leftarrow$ Discount factor for Discounted UCB
6:     $\beta \leftarrow$ Scaling factor for exploration bonuses
7:     $X_i \leftarrow$ Discounted sum of rewards for task $i$
8:     $\bar{X}_i \leftarrow$ Mean of discounted sum of rewards for task $i$
9:     $c_i \leftarrow$ Exploration bonus for task $i$
10:     $n_i \leftarrow$ Discounted count of the number of episodes the agent has trained on task $i$
11:     amta $\leftarrow$ The Active Sampling multi-tasking agent
12:     $X_i \leftarrow 0 \ \forall i$
13:     $n_i \leftarrow 0 \ \forall i$
14:     $\bar{X}_i \leftarrow 0 \ \forall i$
15:     $\bar{h}_i \leftarrow 1 \ \forall i$
16:     **for** train_steps:0 **to** $t$ **do**
17:         $j \leftarrow \mathrm{argmax}_l \ \bar{X}_l + \beta c_l$     // The next task on which the multi-tasking agent is trained
18:         score $\leftarrow$ amta.train_for_one_episode($T_j$)
19:         **if** score $\geq h_j$ **then**
20:             $h_j \leftarrow 2 \times h_j$
21:         $X_i \leftarrow \gamma X_i \ \forall i$
22:         $X_j \leftarrow X_j + \max\left(\frac{h_j - score}{h_j}, 0\right)$
23:         $n_i \leftarrow \gamma n_i \ \forall i$
24:         $n_j \leftarrow n_j + 1$
25:         $\bar{X}_i \leftarrow X_i / n_i \ \forall i$
26:         $c_i \leftarrow \sqrt{\frac{\max(\bar{X}_i(1 - \bar{X}_i), 0.002) \times \log(\sum_l n_l)}{n_i}} \ \forall i$

---

## APPENDIX D: TRAINING CURVES FOR ALL METHODS AND MULTITASKING INSTANCES

This appendix contains the evolution of the raw game-play performance (measured using an evaluation procedure mentioned in Appendix B) versus training progress. STA3C refers to the performance of a single-task A3C agent (Mnih et al., 2016a) trained on a particular task.

### MULTI-TASKING INSTANCE 1 (MT1)

This multi-tasking instance has 6 tasks.

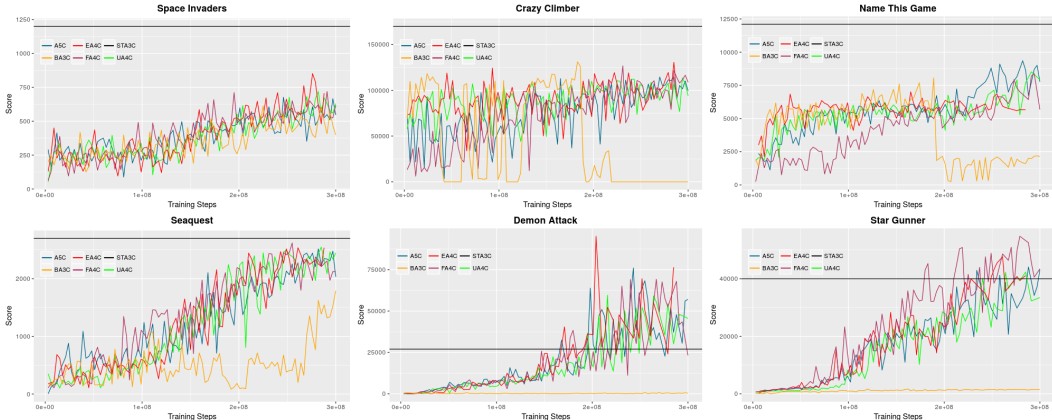

Figure 7: Comparison of performance of BA3C, A5C, UA4C, EA4C and FA4C agents along with task-specific A3C agents for MT1 (6 tasks). Agents in these experiments were trained for 300 million time steps and required half the data and computation that would be required to train the task-specific agents (STA3C) for all the tasks.

### MULTI-TASKING INSTANCE 2 (MT2)

This multi-tasking instance has 6 tasks as well.

### MULTI-TASKING INSTANCE 3 (MT3)

This multi-tasking instance has 6 tasks as well.

### MULTI-TASKING INSTANCE 4 (MT4)

This multi-tasking instance has 8 tasks. This is the same set of 8 tasks that were used in (Parisotto et al., 2016).

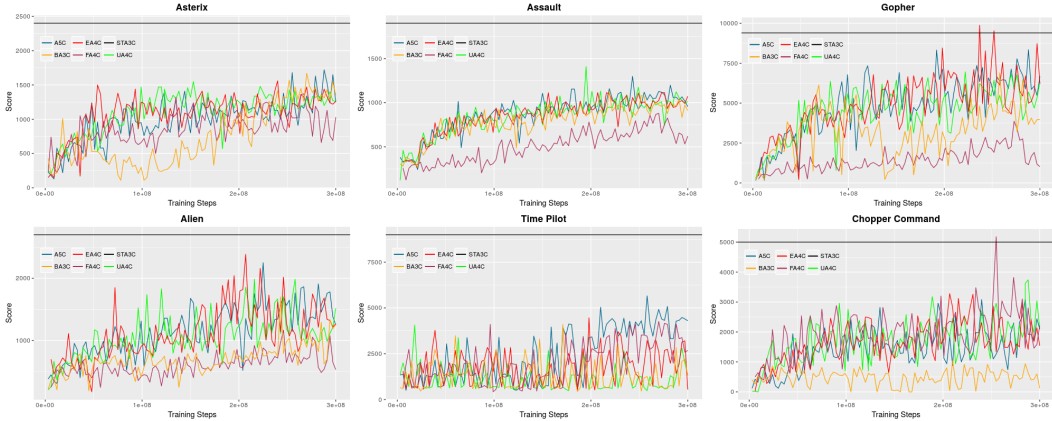

Figure 8: Comparison of performance of BA3C, A5C, UA4C, EA4C and FA4C agents along with task-specific A3C agents for MT2 (6 tasks). Agents in these experiments were trained for 300 million time steps and required half the data and computation that would be required to train the task-specific agents (STA3C) for all the tasks.

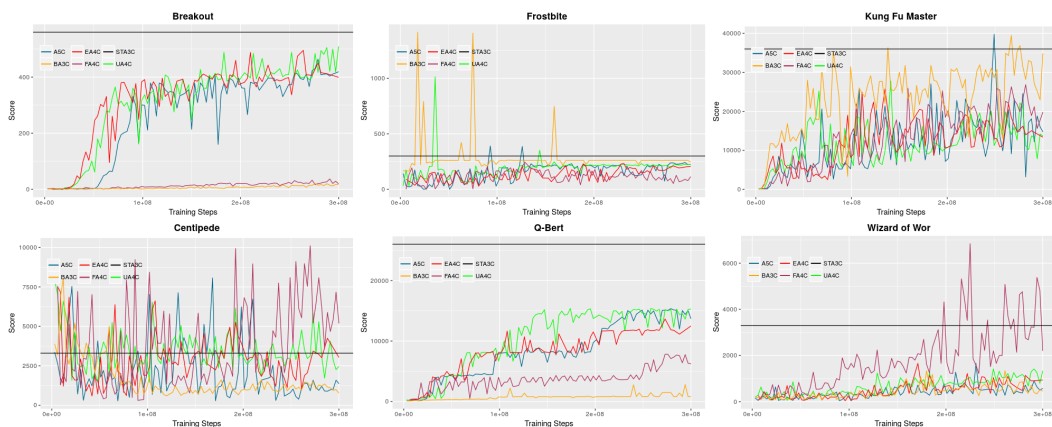

Figure 9: Comparison of performance of BA3C, A5C, UA4C, EA4C and FA4C agents along with task-specific A3C agents for MT3 (6 tasks). Agents in these experiments were trained for 300 million time steps and required half the data and computation that would be required to train the task-specific agents (STA3C) for all the tasks.

MULTI-TASKING INSTANCE 5 (MT5)

This multi-tasking instance has 12 tasks. Although this set of tasks is medium-sized, the multi-tasking network has the same size as those used for MT1, MT2 and MT3 as well as a single-task network.

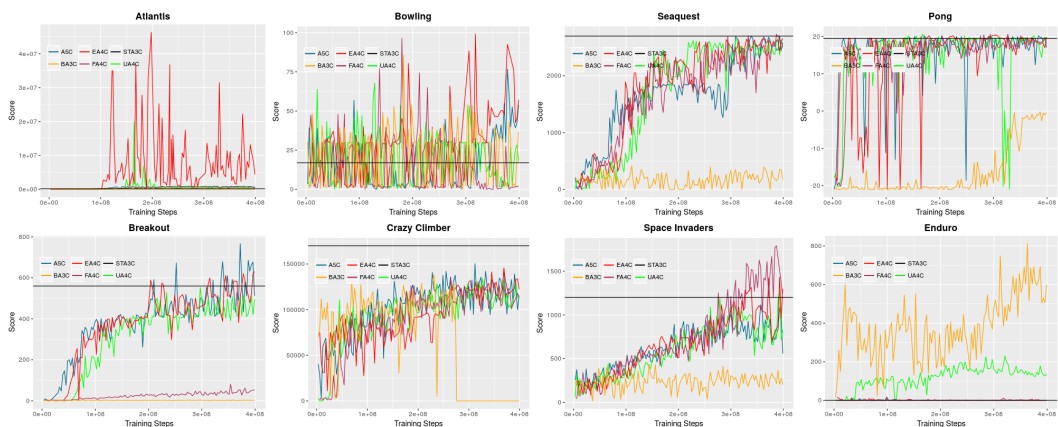

Figure 10: Comparison of performance of BA3C, A5C, UA4C, EA4C and FA4C agents along with task-specific A3C agents for MT4 (8 tasks). Agents in these experiments were trained for 400 million time steps and required half the data and computation that would be required to train the task-specific agents (STA3C) for all the tasks.

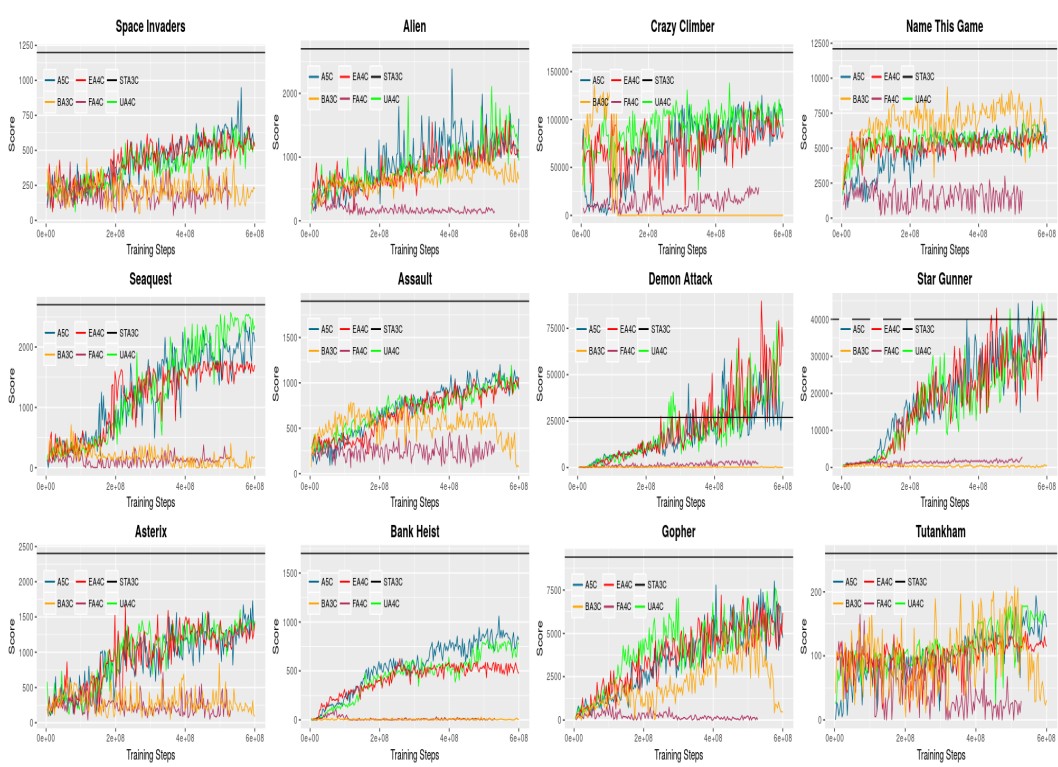

Figure 11: Comparison of performance of BA3C, A5C, UA4C, EA4C and FA4C agents along with task-specific A3C agents for MT5 (12 tasks). Agents in these experiments were trained for 600 million time steps and required half the data and computation that would be required to train the task-specific agents (STA3C) for all the tasks.

MULTI-TASKING INSTANCE 6 (MT6)

This multi-tasking instance has 12 tasks as well.

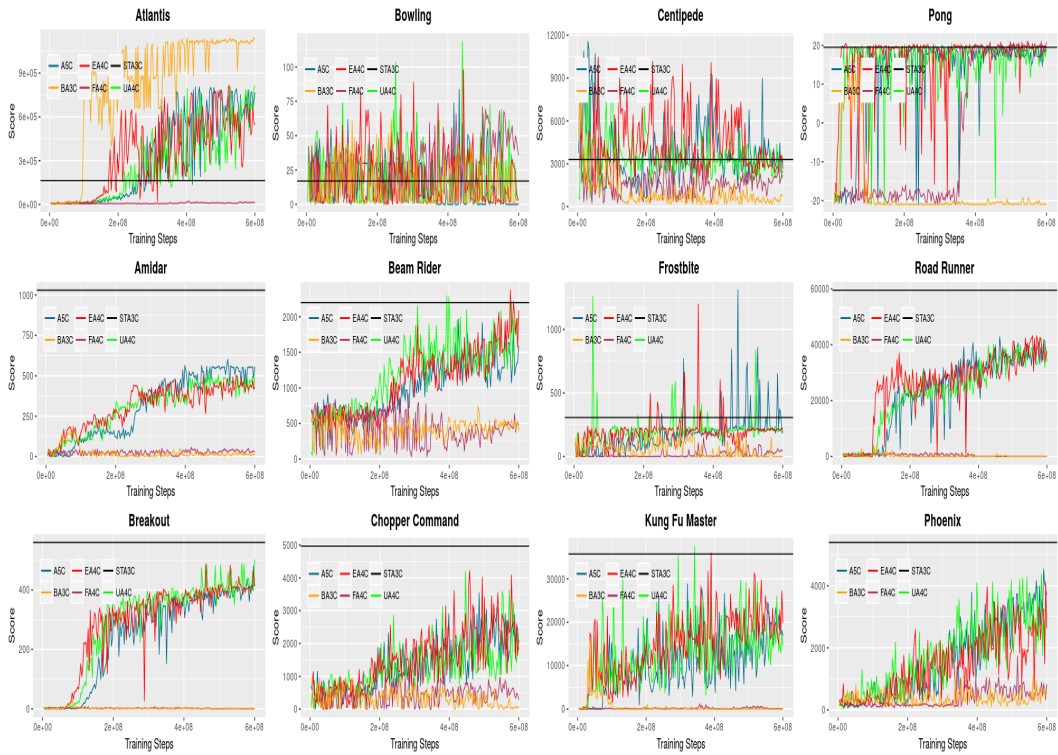

Figure 12: Comparison of performance of BA3C, A5C, UA4C, EA4C and FA4C agents along with task-specific A3C agents for MT6 (12 tasks). Agents in these experiments were trained for 600 million time steps and required half the data and computation that would be required to train the task-specific agents (STA3C) for all the tasks.

MULTI-TASKING INSTANCE 7 (MT7)

This multi-tasking instance has 21 tasks. This is a large-sized set of tasks. Since a single network now needs to learn the prowess of 21 visually different Atari tasks, we roughly doubled the number of parameters in the network, compared to the networks used for MT1, MT2 and MT3 as well as a single-task network. We believe that this is a fairer large-scale experiment than those done in (Rusu et al., 2016a) wherein for a multi-tasking instance with 10 tasks, a network which has **four** times as many parameters as a single-task network is used.

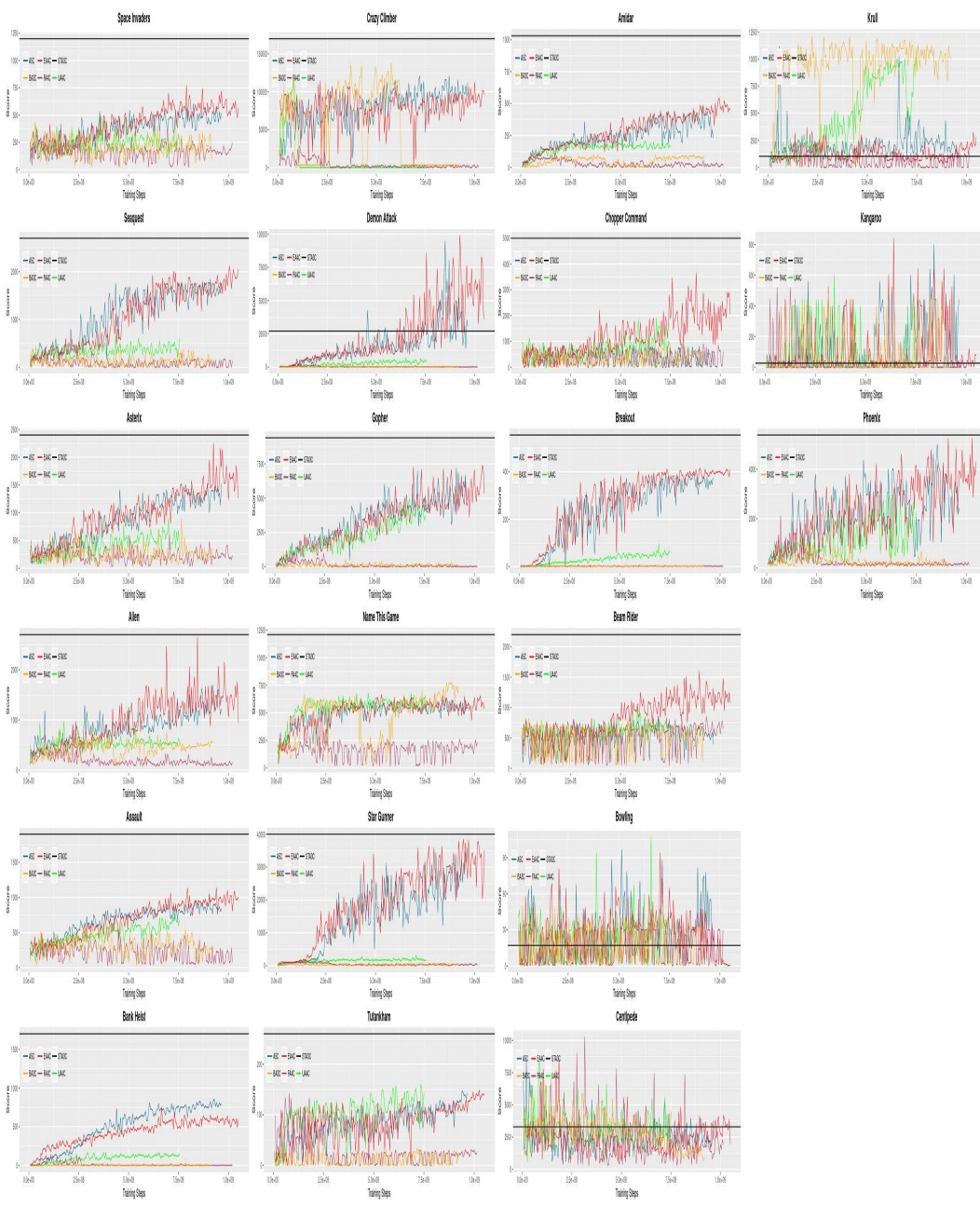

Figure 13: Comparison of performance of BA3C, A5C, UA4C, EA4C and FA4C agents along with task-specific A3C agents for MT7 (21 tasks). Agents in these experiments were trained for 1.05 billion time steps and required half the data and computation that would be required to train the task-specific agents (STA3C) for all the tasks.

## APPENDIX E: PERFORMANCE OF OUR METHODS ON ALL THE EVALUATION METRICS

In this appendix, we document the performance of our methods on the all the four performance metrics ($p_{\mathrm{am}}, q_{\mathrm{am}}, q_{\mathrm{gm}}, q_{\mathrm{hm}}$) that have been proposed in Section 4.1.

$q_{am}$ is a robust evaluation metric because the agent needs to be good in all the tasks in order to get a high score on this metric. In Table 4 we can observe a few important trends:

1. The adaptive method is a hard baseline to beat. The very fact that tasks are being sampled in accordance with the lack of performance of the multi-tasking on them, means that the MTA benefits directly from such a strategy.

2. The UCB-based meta-learner generalizes fairly well to medium-sized instances but fails to generalize to the largest of our multi-tasking instances: MT7.

3. It is our meta-learning method EA4C which generalizes the best to the largest multi-tasking instance MT7. This could be because the UCB and adaptive controllers are more rigid compared to the learning method.

Table 4: Comparison of the performance of our agents based on $q_{am}$

|        | MT1   | MT2   | MT3   | MT4   | MT5   | MT6   | MT7   |
|--------|-------|-------|-------|-------|-------|-------|-------|
| **|T|** | 6    | 6     | 6     | 8     | 12    | 12    | 21    |
| **A5C**  | **0.799** | **0.601** | 0.646 | 0.915 | 0.650 | 0.741 | 0.607 |
| **UA4C** | 0.779 | 0.506 | **0.685** | 0.938 | **0.673** | **0.756** | 0.375 |
| **EA4C** | 0.779 | 0.585 | 0.591 | **0.963** | 0.587 | 0.730 | **0.648** |
| **FA4C** | 0.795 | 0.463 | 0.551 | 0.822 | 0.128 | 0.294 | 0.287 |
| **BA3C** | 0.316 | 0.398 | 0.337 | 0.295 | 0.260 | 0.286 | 0.308 |

Table 5 demonstrates the need for the evaluation metrics that we have proposed. specifically, it can be seen that in case of MT4, the non-clipped average performance is best for BA3C. However, this method is certainly not a good MTL algorithm. This happens because the uniform sampling ensures that the agent trains on the task of Enduro a lot (can be seen in the corresponding training curves). Owing to high performance on a single task, $p_{am}$ ends up concluding that BA3C is the best multi-tasking network.

Table 5: Comparison of the performance of our agents based on $p_{am}$

|        | MT1   | MT2   | MT3   | MT4     | MT5   | MT6   | MT7   |
|--------|-------|-------|-------|---------|-------|-------|-------|
| **|T|** | 6    | 6     | 6     | 8       | 12    | 12    | 21    |
| **A5C**  | 0.978 | **0.601** | 0.819 | 3.506   | 0.733 | 1.312 | **2.030** |
| **UA4C** | 0.942 | 0.506 | 0.685 | 38.242  | **0.810** | **1.451** | 1.409 |
| **EA4C** | **1.079** | 0.585 | 0.658 | 36.085  | 0.743 | 1.231 | 2.010 |
| **FA4C** | 1.054 | 0.470 | **0.892** | 3.067   | 0.128 | 0.536 | 1.213 |
| **BA3C** | 0.316 | 0.398 | 0.669 | **132.003** | 0.260 | 0.286 | 1.643 |

We defined the $q_{gm}$ and $q_{hm}$ metrics because in some sense, the $q_{am}$ metric can still get away with being good on only a few tasks and not performing well on all the tasks. In this limited sense, $q_{hm}$ is probably the best choice of metric to understand the multi-tasking performance of an agent. We can observe that while A5C performance was slightly better than EA4C performance for MT4 according to the $q_{am}$ metric, the agents are much more head to head as evaluated by the $q_{hm}$ metric.

Table 6: Comparison of the performance of our agents based on $q_{gm}$

|  | MT1 | MT2 | MT3 | MT4 | MT5 | MT6 | MT7 |
|---|---|---|---|---|---|---|---|
| **\|T\|** | 6 | 6 | 6 | 8 | 12 | 12 | 21 |
| **A5C** | **0.782** | **0.580** | 0.617 | 0.902 | 0.633 | **0.870** | 0.547 |
| **UA4C** | 0.749 | 0.466 | **0.649** | 0.934 | **0.650** | 0.733 | 0.144 |
| **EA4C** | 0.753 | 0.565 | 0.561 | **0.958** | 0.553 | 0.697 | **0.616** |
| **FA4C** | 0.777 | 0.418 | 0.380 | 0.722 | 0.091 | 0.071 | 0.133 |
| **BA3C** | 0.151 | 0.343 | 0.047 | 0.345 | 0.125 | 0.113 | 0.097 |

Table 7: Comparison of the performance of our agents based on $q_{hm}$

|  | MT1 | MT2 | MT3 | MT4 | MT5 | MT6 | MT7 |
|---|---|---|---|---|---|---|---|
| **\|T\|** | 6 | 6 | 6 | 8 | 12 | 12 | 21 |
| **A5C** | **0.678** | **0.515** | 0.536 | 0.798 | 0.590 | 0.650 | 0.480 |
| **UA4C** | 0.641 | 0.420 | **0.553** | 0.833 | **0.602** | **0.670** | 0.059 |
| **EA4C** | 0.647 | 0.505 | 0.491 | **0.851** | 0.502 | 0.628 | **0.576** |
| **FA4C** | 0.675 | 0.381 | 0.215 | 0.508 | 0.070 | 0.016 | 0.052 |
| **BA3C** | 0.060 | 0.297 | 0.008 | 7.99E-7 | 0.038 | 0.028 | 0.023 |

## APPENDIX F: DEMONSTRATIVE VISUALIZATION OF OUR METHODS

Figure 1 demonstrates a general visual depiction of our method. This appendix contains visualizations specific to each of the methods.

A5C

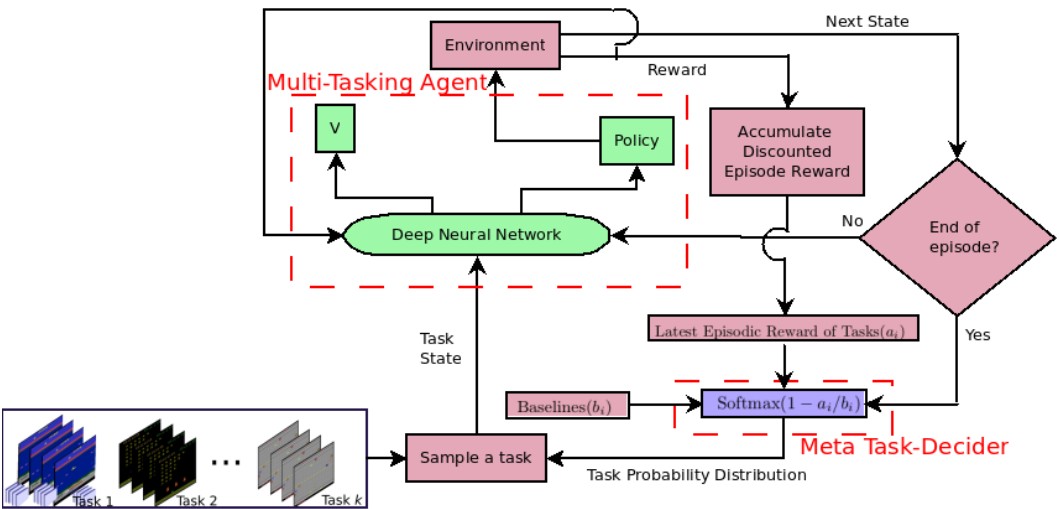

Figure 14: Visual depiction of A5C

UA4C

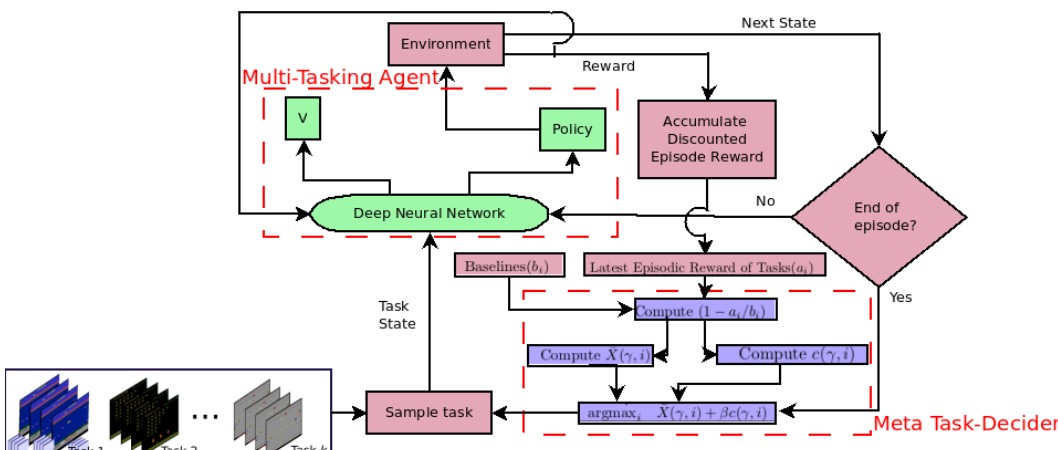

Figure 15: A Visual depiction of UA4C

EA4C

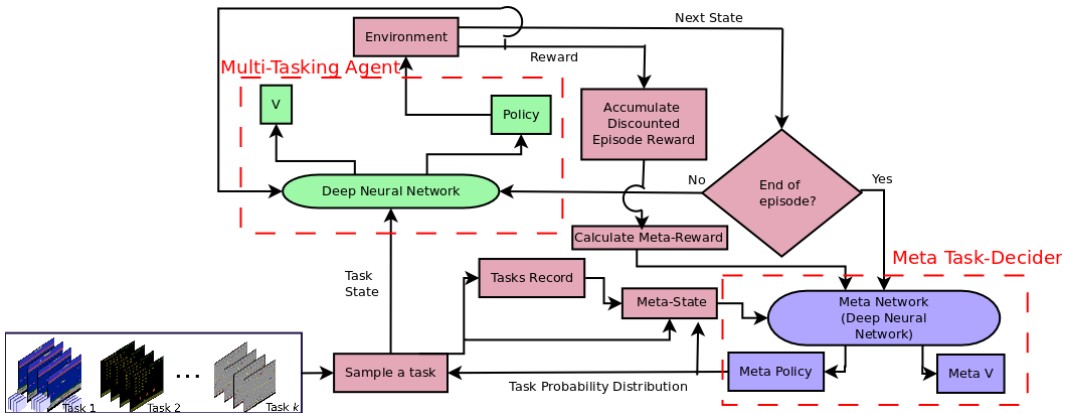

Figure 16: A Visual depiction of EA4C

FA4C

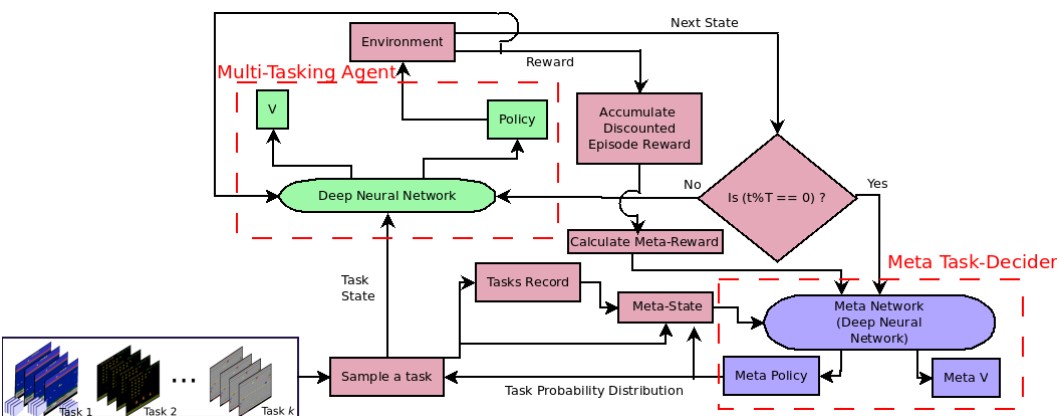

Figure 17: A Visual depiction of FA4C

## APPENDIX G: ROBUSTNESS OF OUR PROPOSED FRAMEWORK (A4C) TO TARGET SCORES

### ON THE ROBUSTNESS OF A4C TO TARGET SCORES

To demonstrate that our framework is robust to the use of different target scores, we performed two targeted experiments.

### USE OF HUMAN SCORES

In this first experiment, we swapped out the use of single-task scores as target scores with the use of scores obtained by Human testers. These human scores were taken from (Mnih et al., 2015). We experimented with UA4C on MT1 in this subsection. Consequently we refer to the use of human scores in UA4C as HUA4C. Figure 18 depicts the evolution of the raw performance (game-score)

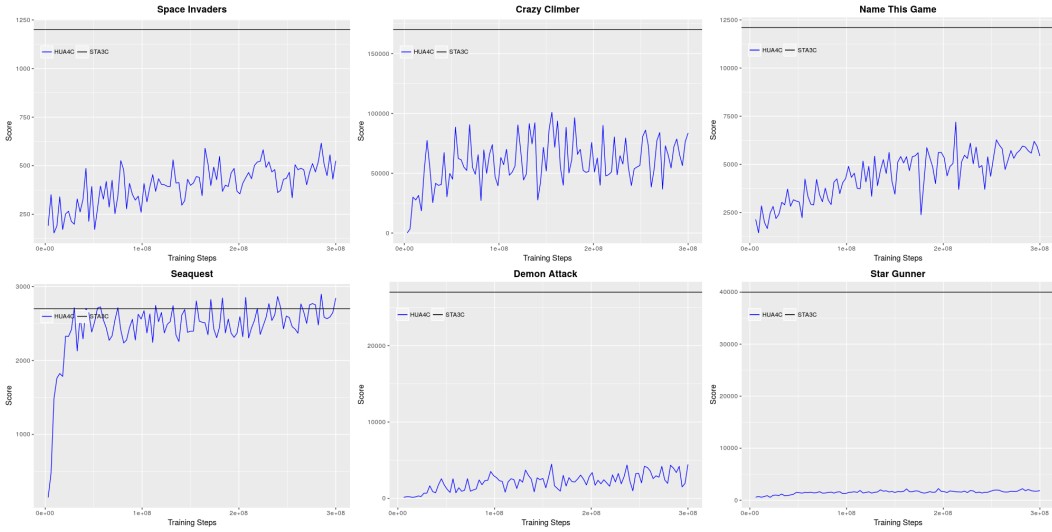

Figure 18: Training curve for HUA4C: when human scores are used as target for calculating the rewards.

of HUA4C agent trained with human scores as targets instead of single-task network's scores. The performance of HUA4C on all the metrics proposed in this paper is contained in Table 8.

Table 8: Evaluation of the HUA4C agent on MT1 using all of our proposed evaluation metrics.

|  | $p_{am}$ | $q_{am}$ | $q_{gm}$ | $q_{hm}$ |
|---:|:---:|:---:|:---:|:---:|
| **HUA4C** | **0.953** | **0.616** | **0.460** | **0.311** |
| **BA3C** | 0.316 | 0.316 | 0.151 | 0.06 |

All the hyper-parameters found by tuning UA4C on MT1 were retained. None of the hyper-parameters were re-tuned. This represents a setup which isn't very favorable for HUA4C. We observe that even in this setup, the performance of UA4C is impressive. However, it is unable to learn at all for two of the tasks and has at best mediocre performance in three others. We believe that the performance could possibly improve if hyper-parameters were tuned for this specific paradigm/framework.

USE OF TWICE THE SINGLE-TASK SCORES AS TARGETS

To demonstrate that the impressive performance of our methods is not conditioned on the use of single-task performance as target scores, we decided to experiment with *twice* the single-task performance as the target scores. In some sense, this *twice the single-task performance* score represents a very optimistic estimate of how well an MTA can perform on a given task. All experiments in this sub-section are performed with A5C. Since the hyper-parameters for all the methods were tuned on $MT_1$, understandably, the performance of our agents is better on $MT_1$ than $MT_2$ or $MT_3$. Hence we picked the multi-tasking instances $MT_2$ and $MT_3$ to demonstrate the effect of using twice the target scores which were used by A5C. We chose the twice-single-task-performance regime arbitrarily and merely wanted to demonstrate that a change in the target scores does not adversely affect our methods' performance. Note that we did not tune the hyper-parameters for experiments in this sub-section. Such a tuning could potentially improve the performance further. It can be seen that in every case, the use of twice-the-single-task-performance as target scores improves the performance of our agents. In some cases such as $MT_3$ there was a large improvement.

Table 9: Comparison of our agents between the case of using usual target scores and double target scores.

| Name | Agent | Target | $q_{am}$ | $q_{gm}$ | $q_{hm}$ |
|------|-------|--------|----------|----------|----------|
| $MT_2$ | A5C | Usual | 0.601 | 0.580 | 0.515 |
| $MT_2$ | A5C | Twice | **0.630** | **0.596** | **0.520** |
| $MT_2$ | BA3C | - | 0.3978 | 0.343 | 0.297 |
| $MT_3$ | A5C | Usual | 0.646 | 0.617 | 0.536 |
| $MT_3$ | A5C | Twice | **0.725** | **0.691** | **0.600** |
| $MT_3$ | BA3C | - | 0.337 | 0.0047 | 0.008 |

APPENDIX H: FIRING ANALYSES FOR ALL MULTITASKING INSTANCES

In this section, we present the results from Firing Analyses done for all the MTIs in this work. The method used to generate the following graphs has been described in section 7.2. It can be seen from the following graphs that the active sampling methods(A5C,UA4C and EA4C) have a large fraction of neurons that fire for a large fraction of time in atleast half the number of tasks in the MTI, whereas BA3C has a relatively higher fraction of task-specific neurons. This alludes to the fact that the active sampling methods have been successful in learning useful features that generalize across different tasks, hence leading to better performance.

**Neuron-Firing Analysis on MT1:**

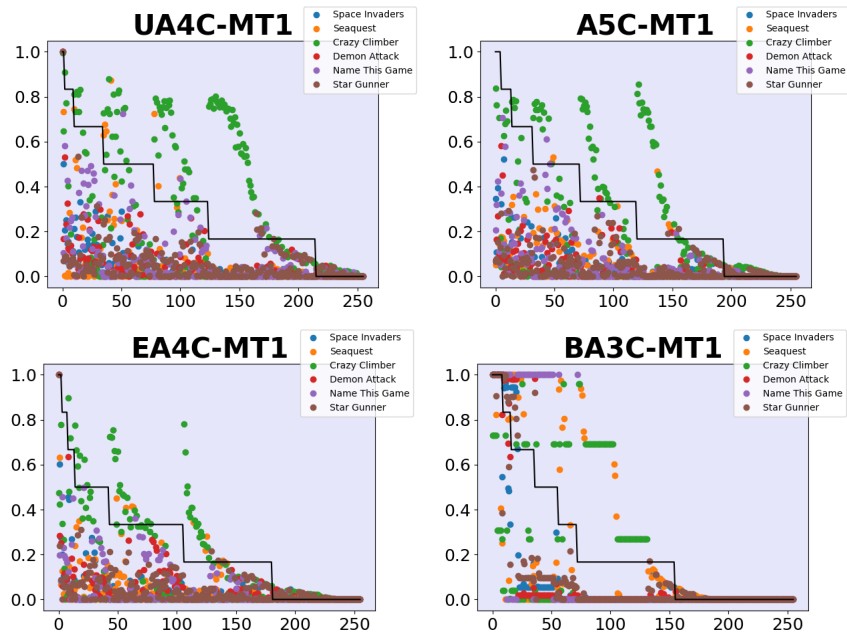

Figure 19: Understanding abstract LSTM features in our proposed methods by analyzing firing patterns on *MT1*

**Neuron-Firing Analysis on MT2:**

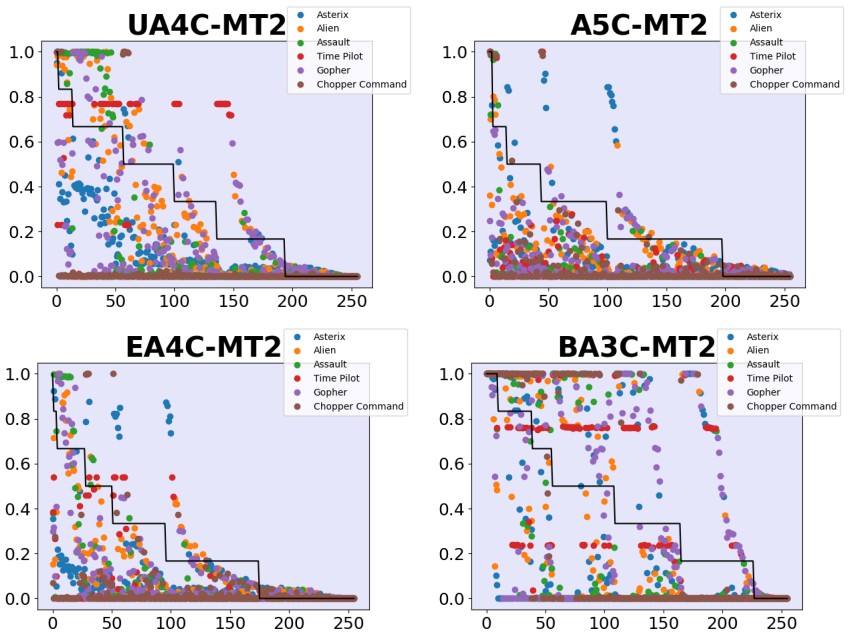

Figure 20: Understanding abstract LSTM features in our proposed methods by analyzing firing patterns on *MT2*

**Neuron-Firing Analysis on MT3:**

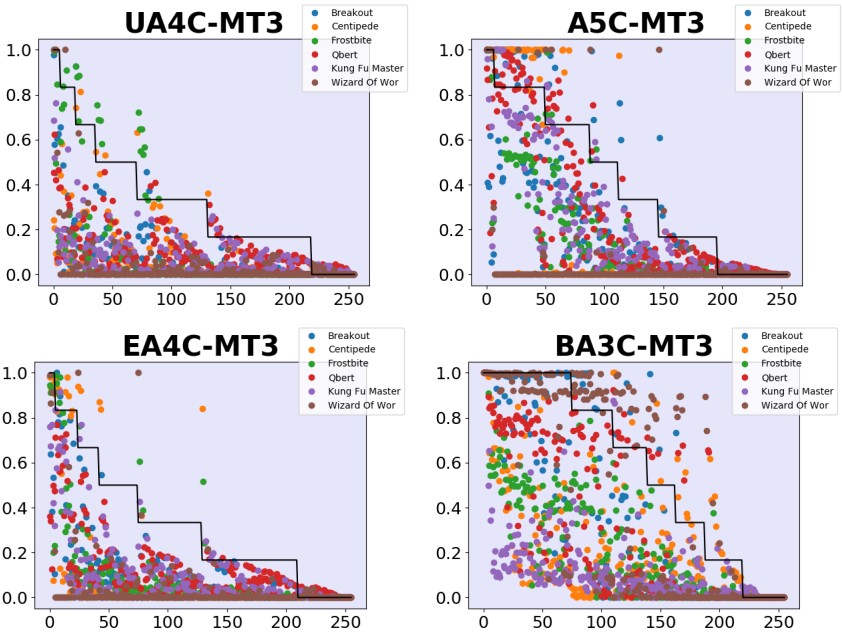

Figure 21: Understanding abstract LSTM features in our proposed methods by analyzing firing patterns on *MT3*

**Neuron-Firing Analysis on MT4:**

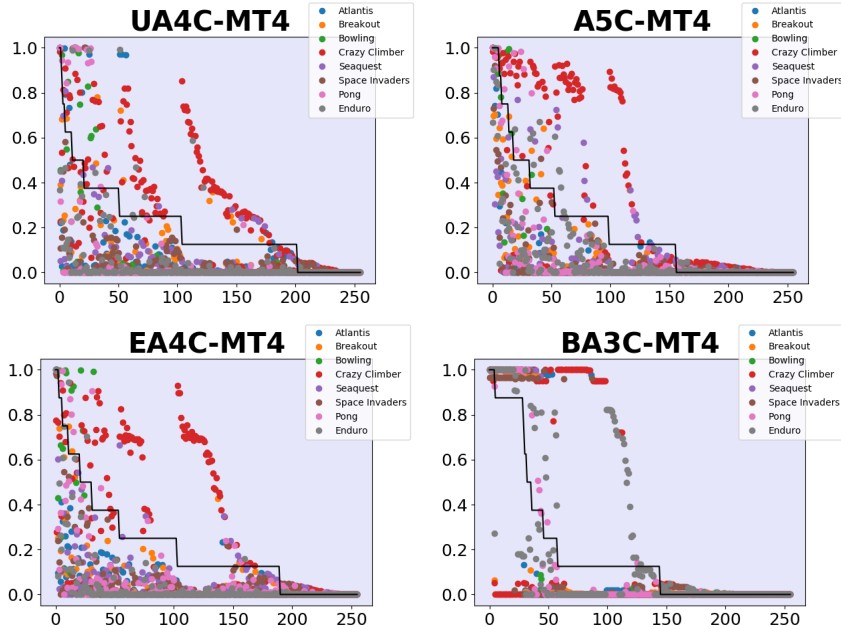

Figure 22: Understanding abstract LSTM features in our proposed methods by analyzing firing patterns on *MT4*

**Neuron-Firing Analysis on MT5:**

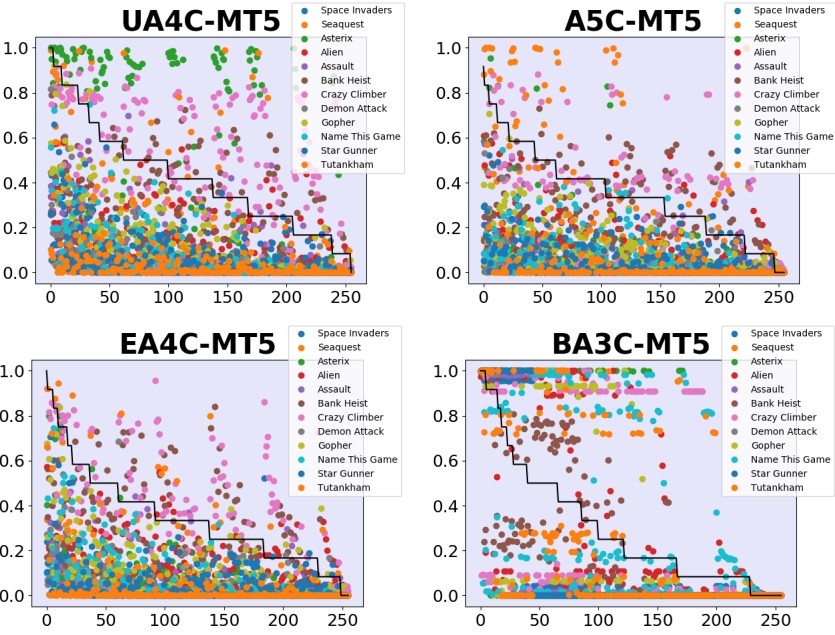

Figure 23: Understanding abstract LSTM features in our proposed methods by analyzing firing patterns on *MT5*

**Neuron-Firing Analysis on MT6:**

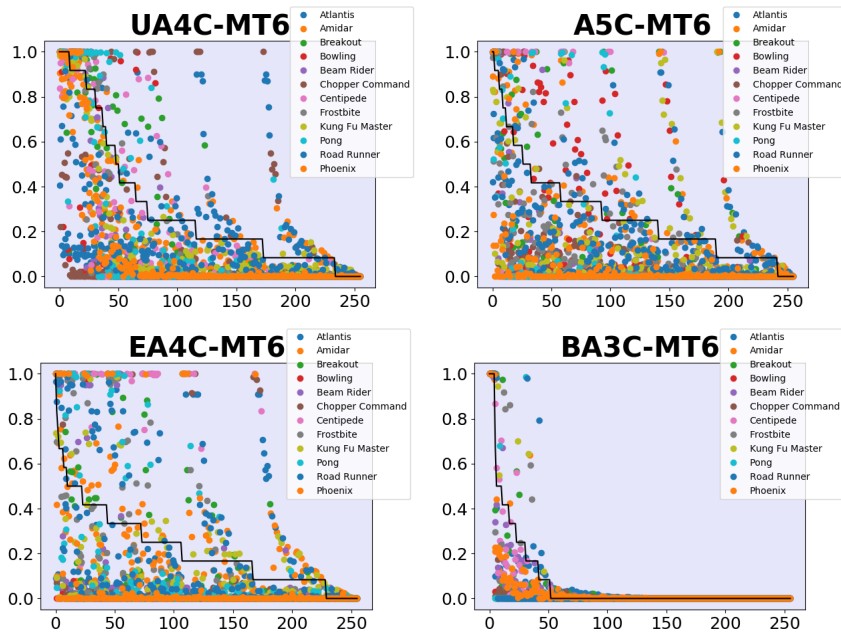

Figure 24: Understanding abstract LSTM features in our proposed methods by analyzing firing patterns on *MT6*

**Neuron-Firing Analysis on MT7:**

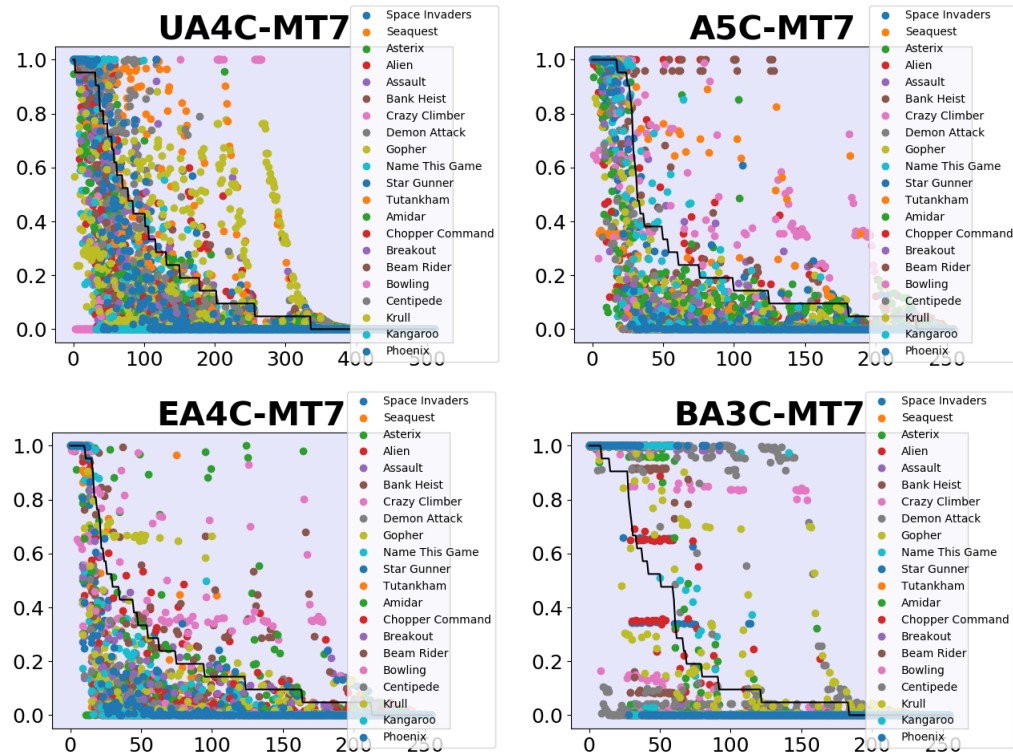

Figure 25: Understanding abstract LSTM features in our proposed methods by analyzing firing patterns on *MT7*

APPENDIX I: DIFFERENT OUTPUT HEADS FOR DIFFERENT TASKS

Some works in multi-task learning (Parisotto et al., 2016) have experimented with baselines wherein all but one layer of the MTN are shared across tasks (to enable the learning of task-agnostic common representation of the different state spaces) but the final layer is task-specific. In fact (Rusu et al., 2016a) used different output heads for their main method, the Policy-distillation based MTA as well. We wanted to understand whether the use of different output heads helps with the problem of multi-tasking or not. The reason that this section is in the appendix because we answer that question in the negative.

This appendix consists of experiments with such an agent which has an output head specific to each task. The method we decided to test was A5C since is the simplest of methods that has impressive performance, specially on the smaller MT instances. This different heads agent is known as a DA5C agent (Different-heads Adaptive Active-sampling A3C). Architectures for DA5C and A5C multi-tasking agents are exactly the same except for what is described below. Suppose that the size of the union of the action spaces of all the tasks is $S$. For Atari games $S = 18$. The only difference in DA5C's architecture with respect to A5C's architecture is that instead of a final $S$-dimensional softmax layer, the $S$ dimensional pre-softmax output vector is projected into $k$ different vector subspaces where $k$ is the number of tasks that DA5C is solving. The projection matrices ($W_i$'s) are akin to a linear activation layer in a neural network and are learned end to end along with the rest of the task. Let the $S$-dimensional pre-softmax output of A4CDH be denoted as $v$. Then the policy for task $i$ is given by:

$$z_i = W_i v$$
$$\pi_{ij} = \frac{e^{z_{ij}}}{\sum_k e^{z_{ik}}}$$

where $\pi_{ij}$ is the probability of picking action $j$ in task $i$. $W_i$ is a projection matrix for task $i$.

The output size for head $h_i$ is the number of valid actions for task $i$. Hence, the total number of extra parameters introduced in DA5C agents (over and above A4CSH agents) is only $S \times \sum_{i=0}^{k-1} |A_i|$ where $A_i$ is the set of valid actions for task $i$. Hyper-parameter tuning similar to A5C was performed on $MT_1$. After hyper-parameter tuning, we found the best hyper-parameters for this agent to be $\beta = 0.02$ and $\tau = 0.05$. Baseline different-head agents (DBA3C) can be defined, similar to the BA3C agents, but with task-specific output heads for each task. The best $\beta$ for DBA3C agents was found to be $\beta = 0.1$.

The performance of different-head agents (DA5C and DBA3C) on all the small multi-tasking instances (MT1, MT2 and MT3) is shown in Table 10. It is clear from Table 10 that DA5C out-performs DBA3C, specially if one would like to optimize for the performance of the

Table 10: Comparison of performance of DA5C agents to DBA3C agents.

| Name | Agent | $p_{am}$ | $q_{am}$ | $q_{gm}$ | $q_{hm}$ |
|---|---|---|---|---|---|
| $MT_1$ | DA5C | **0.907** | **0.771** | **0.739** | **0.630** |
| $MT_1$ | DBA3C | 0.244 | 0.244 | 0.130 | 0.063 |
| $MT_2$ | DA5C | **0.525** | **0.525** | **0.511** | **0.460** |
| $MT_2$ | DBA3C | 0.302 | 0.302 | 0.094 | 0.026 |
| $MT_3$ | DA5C | 0.452 | **0.385** | **0.187** | **0.033** |
| $MT_3$ | DBA3C | **0.471** | 0.339 | 0.048 | 0.008 |

multi-tasking agent on all the tasks. However, it is to be noted that the shared-head versions (A5C) strictly out-perform their different-head counterparts (DA5C). Using this appendix as empirical evidence, we argue against the use of different head-based multi-tasking agents as presented in (Rusu et al., 2016a), specially when the constituent tasks inherently have a common action space.

APPENDIX J: TARGET SCORES FOR ALL THE TASKS

This section contains the list of target scores for all the tasks used in the experiments in our work. These were taken as the single task A3C scores from Table 4 of (Sharma et al., 2017).

Table 11: Table of target scores used for the tasks

| Task | Target |
|---|---|
| Space Invaders | 1200 |
| Seaquest | 2700 |
| Asterix | 2400 |
| Alien | 2700 |
| Assault | 1900 |
| Time Pilot | 9000 |
| Bank Heist | 1700 |
| Crazy Climber | 170000 |
| Demon Attack | 27000 |
| Gopher | 9400 |
| Name This Game | 12100 |
| Star Gunner | 40000 |
| TutanKham | 260 |
| Amidar | 1030 |
| Chopper Command | 4970 |
| Breakout | 560 |
| Beam Rider | 2200 |
| Bowling | 17 |
| Centipede | 3300 |
| Krull | 1025 |
| Kangaroo | 26 |
| Phoenix | 5384 |
| Atlantis | 163660 |
| Frostbite | 300 |
| Kung Fu Master | 36000 |
| Pond | 19.5 |
| Road Runner | 59540 |
| Qbert | 26000 |
| Wizard of Wor | 3300 |
| Enduro | 0.77 |

## APPENDIX K: ADDITIONAL RESULTS ON DOUBLING PARADIGM

RESULTS ON ALL THE METRICS FOR DIFFERENT MTIS

Table 12: Comparison of the performance DUA4C agent to BA3C based on $q_{am}$

|        | MT1   | MT2   | MT4   | MT5   |
|--------|-------|-------|-------|-------|
| **|T|** | 6     | 6     | 8     | 12    |
| **DUA4C** | 0.661 | 0.533 | 0.576 | 0.509 |
| **BA3C** | 0.316 | 0.398 | 0.295 | 0.260 |

Table 13: Comparison of the performance DUA4C agent to BA3C based on $p_{am}$

|        | MT1   | MT2   | MT4     | MT5   |
|--------|-------|-------|---------|-------|
| **|T|** | 6     | 6     | 8       | 12    |
| **DUA4C** | 0.739 | 0.549 | 134.939 | 0.577 |
| **BA3C** | 0.316 | 0.398 | 132.003 | 0.260 |

Table 14: Comparison of the performance DUA4C agent to BA3C based on $q_{gm}$

|        | MT1   | MT2   | MT4   | MT5   |
|--------|-------|-------|-------|-------|
| **|T|** | 6     | 6     | 8     | 12    |
| **DUA4C** | 0.452 | 0.463 | 0.398 | 0.381 |
| **BA3C** | 0.151 | 0.343 | 0.345 | 0.125 |

Table 15: Comparison of the performance DUA4C agent to BA3C based on $q_{hm}$

|        | MT1   | MT2   | MT4     | MT5   |
|--------|-------|-------|---------|-------|
| **|T|** | 6     | 6     | 8       | 12    |
| **DUA4C** | 0.175 | 0.371 | 0.217   | 0.218 |
| **BA3C** | 0.060 | 0.297 | 7.99E-7 | 0.038 |

TRAINING CURVES FOR DIFFERENT MTIS

MULTI-TASKING INSTANCE 1 (MT1)

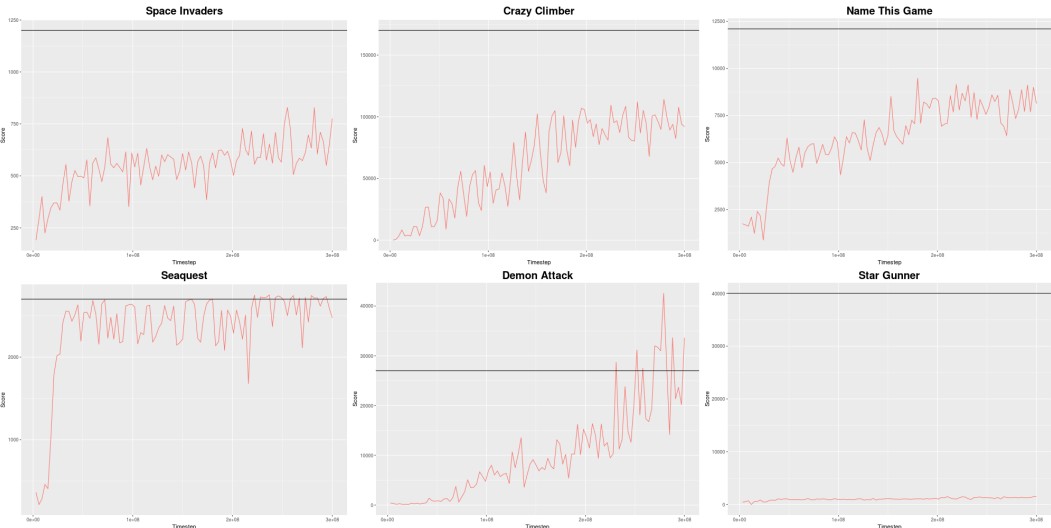

Figure 26: Training Curves for the DUA4C agent on MT1 (6 tasks). The horizontal line represents the Single Task Agent's score. Agents in these experiments were trained for 300 million time steps and required half the data and computation that would be required to train the task-specific agents (STA3C) for all the tasks.

MULTI-TASKING INSTANCE 2 (MT2)

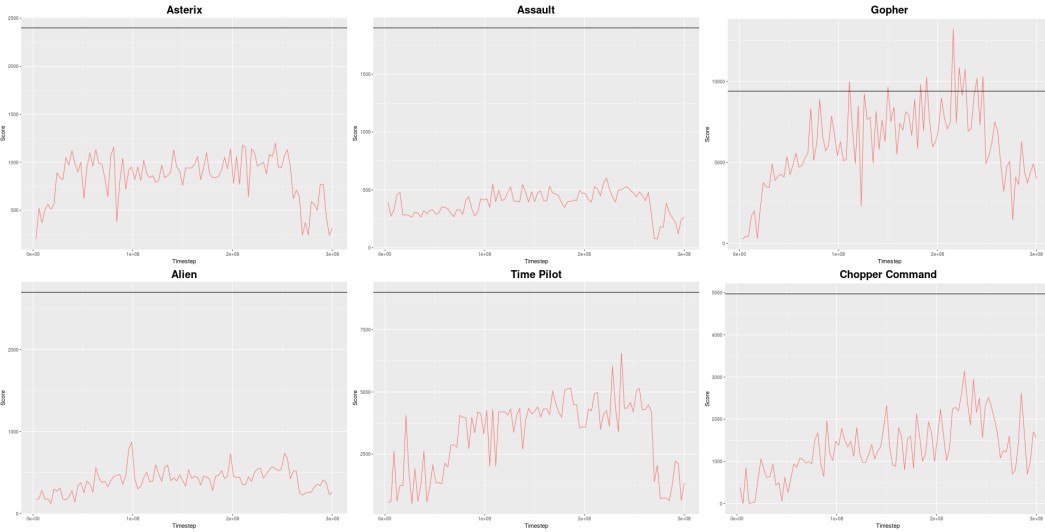

Figure 27: Training Curves for the DUA4C agent on MT2 (6 tasks). The horizontal line represents the Single Task Agent's score. Agents in these experiments were trained for 300 million time steps and required half the data and computation that would be required to train the task-specific agents (STA3C) for all the tasks.

MULTI-TASKING INSTANCE 4 (MT4)

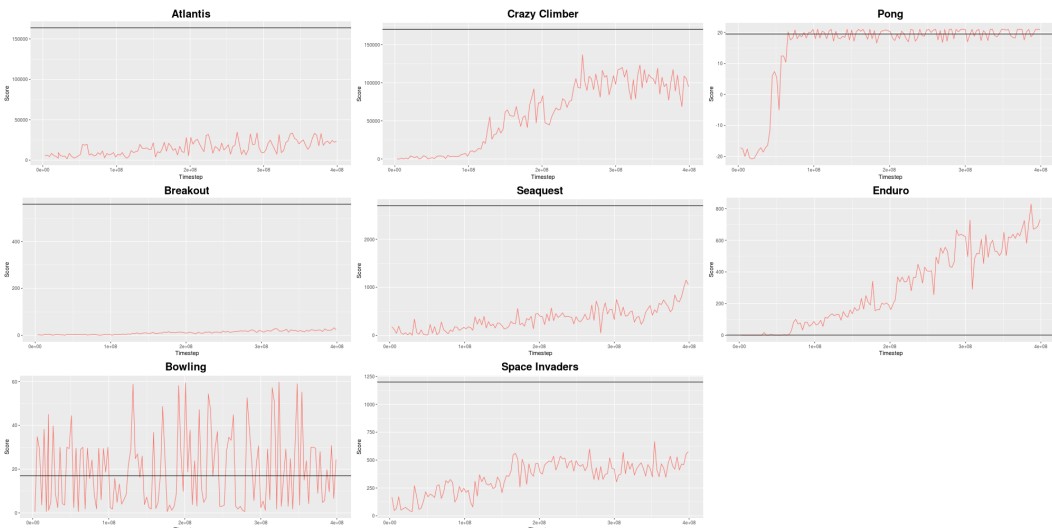

Figure 28: Training Curves for the DUA4C agent on MT4 (8 tasks). The horizontal line represents the Single Task Agent's score. Agents in these experiments were trained for 400 million time steps and required half the data and computation that would be required to train the task-specific agents (STA3C) for all the tasks.

MULTI-TASKING INSTANCE 5 (MT5)

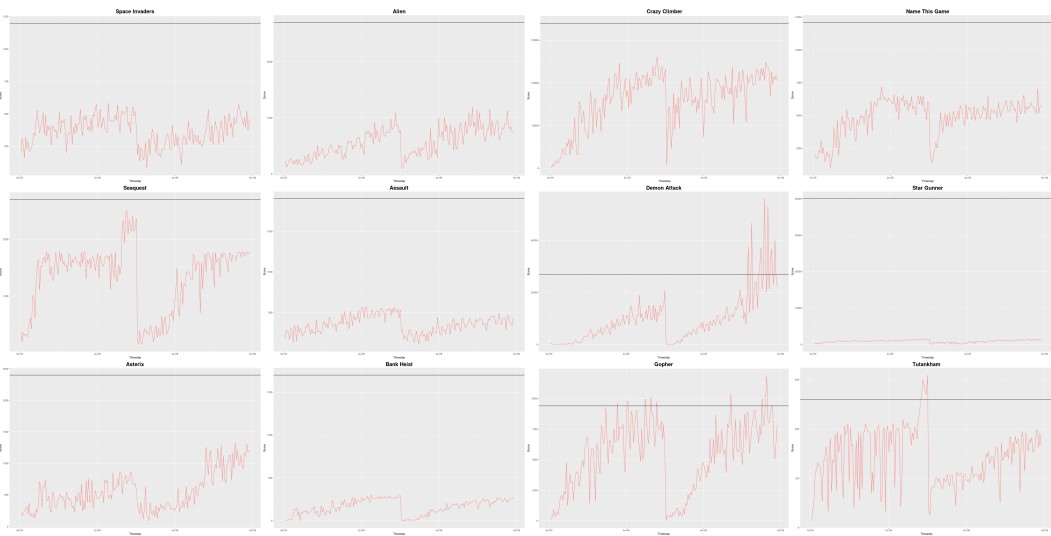

Figure 29: Training Curves for the DUA4C agent on MT5 (12 tasks). The horizontal line represents the Single Task Agent's score. Agents in these experiments were trained for 600 million time steps and required half the data and computation that would be required to train the task-specific agents (STA3C) for all the tasks.

