# OpenReview forum: "Learning to Multi-Task by Active Sampling"
_ICLR.cc/2018/Conference — Accept (Poster)_

### Official Review · AnonReviewer3 · 2017-11-27
**New online algorithms for learning multiple sequential problems**

**Rating:** 5
**Confidence:** 3

**Review:**

The paper present online algorithms for learning multiple sequential problems. The main contribution is to introduce active learning principles for sampling the sequential tasks in an online algorithm. Experimental results are given on different multi-task instances. The contributions are interesting and experimental results seem promising. But the paper is difficult to read due to many different ideas and because some algorithms and many important explanations must be found in the Appendix (ten sections in the Appendix and 28 pages). Also, most of the paper is devoted to the study of algorithms for which the expected target scores are known. This is a very strong assumption. In my opinion, the authors should have put the focus on the DU4AC algorithm which get rids of this assumption. Therefore, I am not convinced that the paper is ready for publication at ICLR'18.
* Differences between BA3C and other algorithms are said to be a consequence of the probability distribution over tasks. The gap is so large that I am not convinced on the fairness of the comparison. For instance, BA3C (Algorithm 2 in Appendix C) does not have the knowledge of the target scores while others heavily rely on this knowledge.
* I do not see how the single output layer is defined.
* As said in the general comments, in my opinion Section 6 should be developped and more experiments should be done with the DUA4C algorithm.
* Section 7.1. It is not clear why degradation does not happen. It seems to be only an experimental fact.

---

> ### Author Response · Authors · 2018-01-05
> **Rebuttal for Reviewer 3**
>
> Thank you for the reviews. We address your comments below:
>
> > In my opinion, the authors should have put the focus on the DU4AC algorithm which get rids of this assumption.
> We believe that the Doubling Paradigm is an important part of the paper and thus, as requested by the reviewer, we have added additional results for the DUA4C agent.
>
> Apart from MT1, we now show results on another 6 task instance (MT2), one 8 task instance (MT4) and one 12 task instance (MT5).
> In all the cases, the DUA4C agent outperforms the BA3C agent and is able to perform well on all the MTIs.
> We are still running the DUA4C agent on the 21 task instance and will be able to add the results on the same in the camera-ready version of the paper. These results have increased the quality of our work and we hope the reviewer raises his score in the light of these new experiments.
>
>
> > Differences between BA3C and other algorithms are said to be a consequence of the probability distribution over tasks. The gap is so large that I am not convinced on the fairness of the comparison. For instance, BA3C (Algorithm 2 in Appendix C) does not have the knowledge of the target scores while others heavily rely on this knowledge.
>
> As stated in Section 4.1, we do believe that the lackluster performance of BA3C agent is due to the uniform sampling of the tasks. The DUA4C agent is not provided with the baselines either and it is nevertheless able to beat the BA3C agent by a margin on all the MTIs. The experiments with DUA4C verify our claim that it is indeed the probability distribution over the tasks that causes the huge improvement in our agents.
>
>
> > I do not see how the single output layer is defined.
>
> As stated in Section 3, the single output layer is a superset of all the actions in different tasks. Take an MTI with Pong and Breakout. Pong has valid actions as up, down, and no-op(do nothing). Breakout has valid actions as left, right and no-op. The single output layer will have valid actions as up, down, left, right and no-op. While playing an episode of Pong, if the agent chooses left or right(non-valid actions for Pong), it would be treated as a no-op action.
> In all our experiments, since we deal with Atari Games, we set the output layer as all the possible 18 actions in ALE with non-valid actions as a no-op.
> You can now see how not providing the identity of the task makes learning hard. The agent on seeing a frame is supposed to figure out what is the valid action subset first and thus, learning is harder.
>
>
> > As said in the general comments, in my opinion Section 6 should be developed and more experiments should be done with the DUA4C algorithm.
>
> We have hopefully addressed the issue of developing DUA4C further with the new experiments.
>
>
> > Section 7.1. It is not clear why degradation does not happen. It seems to be only an experimental fact.
>
> While we do agree that we haven’t provided with a theoretical explanation of why degradation doesn’t happen, Section 7.1 does provide with an intuition for why the algorithm is able to prevent catastrophic forgetting. We reiterate: Catastrophic Forgetting in our agents is avoided due to the way in which we sample the tasks. The probability of a task getting sampled in our agents is higher for those tasks on which the agent is currently bad at. Once the agent becomes good on a task, if degradation has happened on a task which was previously good, the agent will switch back to the other task and will thus ensure that it trains more on the degraded task.

---

> > ### Comment · AnonReviewer3 · 2018-01-09
> > **Comments on rebuttal for reviewer 3**
> >
> > I will slightly increase my score and I will not argue against the paper because the paper contains interesting material. Nevertheless, in my opinion, the active strategy heavily relies on the knowledge of target scores. The paper should have contained a precise description of the DUA4C algorithm --not only experimental results--. For instance, when a target score is doubled and becomes unfeasible, how the algorithm behaves ? Why is there no degradation in this case ?

---

> > > ### Author Response · Authors · 2018-01-29
> > > **Reply to Comments on Rebuttal**
> > >
> > > We thank the reviewer of increasing his score. We further address your comments below:
> > >
> > > > The paper should have contained a precise description of the DUA4C algorithm --not only experimental results--.
> > >
> > > The paper does contain a precise description of the DUA4C algorithm. The Algorithm 7 on Page 23 is exactly that.
> > >
> > > > For instance, when a target score is doubled and becomes unfeasible, how the algorithm behaves ? Why is there no degradation in this case ?
> > >
> > > In our experiments with ALE, we never entered a situation where the estimated target scores became unfeasible. While we do not have empirical evidence to no degradation in this case, we do believe that degradation will still not occur if the reward function as shown in Eq (2) is used. This is because the second half of the reward function tries to make sure that the performance on worst 3 games is good. As a result, once an unfeasible target is set for one of the games, the agent will switch to other games to make sure that not just the worst but the worst 3 games have good performance.
> > >
> > > As a side note, for feasible targets, we have seen that the algorithms we have proposed are robust to the higher than usual target scores as seen in the second half of Appendix G.
> > > As we have stated in the paper, we believe that our work is an important first step in the direction of achieving Multi-Tasking agents using Active Learning principles. We agree with the reviewer that evidence of no degradation in case of unfeasible targets is an interesting addition to the paper and have left the empirical verification of the same to future work.

---

### Official Review · AnonReviewer2 · 2017-11-27
**An active learning approach to multitask RL with significant potential**

**Rating:** 7
**Confidence:** 3

**Review:**


The authors show empirically that formulating multitask RL itself as an active learning and ultimately as an RL problem can be very fruitful.  They design and explore several approaches  to the active learning (or active sampling) problem, from a basic
change to the distribution to UCB to feature-based neural-network based RL. The domain is video games.   All proposed approaches beat the uniform sampling baselines and the more sophisticated approaches do better in the scenarios with more tasks (one multitask  problem had 21 tasks).


Pros:

- very promising results with an interesting active learning approach to multitask RL

- a number of approaches developed for the basic idea

- a variety of experiments, on challenging multiple task problems (up to 21 tasks/games)

- paper is overall well written/clear

Cons:

- Comparison only to a very basic baseline (i.e. uniform sampling)
Couldn't comparisons be made, in some way, to other multitask work?



Additional  comments:

- The assumption of the availability of a target score goes against
the motivation that one need not learn individual networks ..  authors
say instead one can use 'published' scores, but that only assumes
someone else has done the work (and furthermore, published it!).

The authors do have a section on eliminating the need by doubling an
estimate for each task) which makes this work more acceptable (shown
for 6 tasks or MT1, compared to baseline uniform sampling).

Clearly there is more to be done here for a future direction (could be
mentioned in future work section).

- The averaging metrics (geometric, harmonic vs arithmetic, whether
  or not to clip max score achieved) are somewhat interesting, but in
  the main paper, I think they are only used in section 6 (seems like
  a waste of space). Consider moving some of the results, on showing
  drawbacks of arithmetic mean with no clipping (table 5 in appendix E), from the appendix to
  the main paper.


- The can be several benefits to multitask learning, in particular
  time and/or space savings in learning new tasks via learning more
  general features. Sections 7.2 and 7.3 on specificity/generality of
  features were interesting.



--> Can the authors show that a trained network (via their multitask
    approached) learns significantly faster on a brand new game
    (that's similar to games already trained on), compared to learning from
    scratch?

--> How does the performance improve/degrade (or the variance), on the
    same set of tasks, if the different multitask instances (MT_i)
    formed a supersets hierarchy, ie if MT_2 contained all the
    tasks/games in MT_1, could training on MT_2 help average
    performance on the games in MT_1 ? Could go either way since the network
   has to allocate resources to learn other games too.  But is there a pattern?



- 'Figure 7.2' in section 7.2 refers to Figure 5.


- Can you motivate/discuss better why not providing the identity of a
  game as an input is an advantage? Why not explore both
  possibilities? what are the pros/cons? (section 3)

---

> ### Author Response · Authors · 2018-01-05
> **Rebuttal for Reviewer 2**
>
> Thanks for reviewing the paper, the comments and questions! We believe addressing these questions will increase the quality of the work, and we will certainly do that.
>
> > Comparison only to a very basic baseline (i.e. uniform sampling). Couldn't comparisons be made, in some way, to other multitask work?
>
> We do make a direct comparison to another multi-task work. As stated in Section 5, the tasks in MT4 (8 task instance) are exactly the same as those used in Actor Mimic Networks (Parisotto et al., 2015). AMNs achieve a q_am of 0.79 while all of our agents achieve a q_am greater than 0.9.
>
>
> > The assumption … future work section).
>
> Before we go ahead, we would like to reiterate that we see the baselines as target scores that we want to achieve on the tasks. As we have shown in Appendix G, it’s not necessary to take them from published works, a human being could try solving a task and set his score as the target as well. Our algorithm is robust to target scores as well as seen in the same Appendix, i.e you could choose (reasonably) bigger targets as well.
>
> We however also believe that the Doubling Paradigm is an important part of the paper and thus, as requested by the reviewer, we have added additional results for the DUA4C agent. Apart from MT1, we now show results on another 6 task instance (MT2), one 8 task instance (MT4) and one 12 task instance (MT5). We are still running the DUA4C agent on the 21 task instance and will be able to add the results on the same in the camera-ready version of the paper. In all the cases, the DUA4C agent outperforms the BA3C agent and is able to perform well on all the MTIs. These results have increased the quality of our work and we thank the reviewer again for raising these requests.
>
>
> > Can the authors show that a trained network (via their multitask approached) learns significantly faster on a brand new game (that's similar to games already trained on), compared to learning from scratch?
>
> The work we have presented focuses specifically on Multi-task learning only and not transfer learning and thus, we didn’t show results on transfer learning. While we haven’t shown explicit results on transfer learning, we STRONGLY believe that it will indeed be the case that the MTAs will learn faster on a new similar game. This is attributed to the fact that all the agents in our work learn task agnostic features (as shown in Section 7) and having learned these features beforehand will speed up training on a similar new task. All in all, we are currently designating transfer learning as future work.
>
>
> > How does the performance improve/degrade (or the variance), on the same set of tasks, if the different multitask instances (MT_i) formed a supersets hierarchy, ie if MT_2 contained all the tasks/games in MT_1, could training on MT_2 help average performance on the games in MT_1 ? Could go either way since the network has to allocate resources to learn other games too.  But is there a pattern?
>
> We do have a supersets hierarchy in the MTIs we’ve chosen. Note that MT1 is a subset of MT5. We see that it is indeed the case that the network has allocated resources to learn other games too. For an A5C agent trained on MT5, the q_am for just the MT1 tasks is 0.697. For the A5C agent trained on MT1, the q_am is 0.799. Please note that the size of the network is same in both the cases. Clearly, the network has allocated some of its representational power to learn the other games. We, however, do not claim this to be a pattern and this forms an interesting direction for further work. We thank the reviewer for this question. This provides further insight into how the network is allocating its resources for multi-tasking.
>
>
> > 'Figure 7.2' in section 7.2 refers to Figure 5.
>
> We apologize for the typo. We’ve fixed it in the revision.
>
>
> > Can you motivate/discuss better why not providing the identity of a game as an input is an advantage? Why not explore both possibilities? what are the pros/cons? (section 3)
>
> Not providing the identity of the game is clearly not an advantage. This is because the agent now has to figure out the subset of actions which make sense for the task (if the actions not valid for a task are chosen, it is treated as a no-op action). It makes the setup harder to solve. The motivation behind doing this is that in real-world problems, the identity of the tasks might not be provided. We point out that in spite of not providing the identity of the tasks, the agents perform quite well on the MTIs.

---

### Official Review · AnonReviewer1 · 2017-11-28
**Improved multitask deep reinforcement learning with active learning**

**Rating:** 7
**Confidence:** 5

**Review:**

In this paper active learning meets a challenging multitask domain: reinforcement learning in diverse Atari 2600 games. A state of the art deep reinforcement learning algorithm (A3C) is used together with three active learning strategies to master multitask problem sets of increasing size, far beyond previously reported works.

Although the choice of problem domain is particular to Atari and reinforcement learning, the empirical observations, especially the difficulty of learning many different policies together, go far beyond the problem instantiations in this paper. Naive multitask learning with deep neural networks fails in many practical cases, as covered in the paper. The one concern I have is perhaps the choice of distinct of Atari games to multitask learn may be almost adversarial, since naive multitask learning struggles in this case; but in practice, the observed interference can appear even with less visually diverse inputs.

Although performance is still reduced compared to single task learning in some cases, this paper delivers an important reference point for future work towards achieving generalist agents, which master diverse tasks and represent complementary behaviours compactly at scale.

I wonder how efficient the approach would be on DM lab tasks, which have much more similar visual inputs, but optimal behaviours are still distinct.

---

> ### Author Response · Authors · 2018-01-05
> **Rebuttal for Reviewer 1**
>
> Thank you for the positive reviews. We address your comments below:
>
> > The choice of distinct of Atari games to multitask learn may be almost adversarial.
>
> We agree with the reviewer that the choice of tasks in our paper could be adversarial because the state spaces are very different visually. This was intentional (we point it out in the caption of Fig 1) with the purpose of raising the standard of the results and strengthens the work presented because, in spite of the state spaces being so visually different, the agents are able to perform very well on all the tasks as the results show.
>
>
> > How efficient would the algorithm be for visually similar tasks?
>
> As we claim in the introduction to Section 7, an ideal MTA performs well due to learning task-agnostic abstract features which help it generalize across multiple tasks. In the case where tasks have visually similar state spaces, finding such features is clearly easier. We thus believe solving visually similar tasks are easier.
> Applying the framework to environments apart from Atari has currently been left as future work because of time and computational constraints.

---

### Public Comment · (anonymous) · 2017-11-22
**Difference in baseline scores**

There seems to be a difference in the baseline scores reported between page 7 and page 8 for A3C/STA3C scores. If I use the baseline scores from page 7 and compare them against DUA4C, I think DUA4C does better in Demon Attack only?

Have you used a different set of parameters to compute the A3C scores for the DUA4C experiments?

---

> ### Author Response · Authors · 2017-11-25
> **Regarding the discrepancy in plots**
>
> We thank the reader for pointing out the error.
>
> There indeed is a discrepancy in the STA3C scores on Page 7 and 8. We checked our results again and we found that the STA3C scores in Figure 3 on Page 7 are correct (they can be verified in Appendix J of the paper). We'll make the corrections in Figure 4 and update the paper once we're allowed to.
>
> As explained in the paper, the STA3C scores are of no importance during the learning of the DUA4C agent. Thus, we checked our results for the final values of the performance metrics (since they do depend on the STA3C agent scores) in case they needed to be changed as well. We found that the performance metrics (p_am, q_am, etc.) were all calculated using the correct baseline scores and thus they are ALL CORRECT. That is the results reports in Table 2 are correct.
>
> While we agree that DUA4C does better on Demon Attack only (and is nearly equal to STA3C on Seaquest), the purpose of a Multi-tasking network is to do reasonably well on ALL the tasks, which might come at the cost of not doing better than the baseline on some tasks. We had included this discussion in the paper in the performance metrics section where we motivated the new performance metric q_am.
>
> Finally, no we didn't use a different set of parameters for the A3C scores. We're not sure of how the error crept only into the plot. We apologize again for the mistake in Figure 4.

---

> > ### Public Comment · (anonymous) · 2017-11-26
> > **Reply to author(s)**
> >
> > Thank you for the confirmation.

---

### Decision · Program_Chairs · 2018-01-29
**ICLR 2018 Conference Acceptance Decision**

**Decision:**

Accept (Poster)

**Comment:**

The paper contains an interesting way to do online multi task learning, by borrowing ideas from active learning and comparing and contrasting a number of ways on the arcade learning environment.  Like the reviewers, I have some concerns about using the target scores and I think more analysis would be needed to see just how robust this method is to the choice/distribution of target scores (the authors mention that things don't break down as long as the scores are "reasonable", but that's not a particularly empirical nor precise statement).

My inclination is to accept the paper, because of the earnest efforts made by the authors in understanding how DUA4C works. However, I do agree that the paper should have a larger focus on that: basically Section 6 should be expanded, and the experiments should be rerun in such a way that the setup for DUA4C is more "favorable" (in terms of hyper-parameter optimization).  If there's any gap between any of the proposed methods and DUA4C, then this would warrant further analysis of course (since it would mean that there's an advantage to using target scores).